

# Synoptic development during the ACLOUD/PASCAL field campaign near Svalbard in spring 2017

Erlend M. Knudsen[1], Bernd Heinold[2], Sandro Dahlke[3,4], Heiko Bozem[5], Susanne Crewell[1],
Georg Heygster[6], Daniel Kunkel[5], Marion Maturilli[3], Mario Mech[1], Annette Rinke[3],
Holger Schmithüsen[7], André Ehrlich[8], Andreas Macke[9], Christof Lüpkes[7], and Manfred Wendisch[8]

[1]Institute for Geophysics and Meteorology, University of Cologne, Albertus-Magnus-Platz, 50923 Köln, Germany
[2]Modelling of Atmospheric Processes, Leibniz Insititue for Tropospheric Research, Permoserstr. 15, 04318 Leipzig, Germany
[3]Alfred Wegener Institute, Helmholtz Centre for Polar and Marine Research, Telegrafenberg A45, D-14473 Potsdam, Germany
[4]Institute of Physics and Astronomy, University of Potsdam, Karl-Liebknecht-Str. 24/25, 14476 Potsdam-Golm, Germany
[5]Institute for Atmospheric Physics, Johannes Gutenberg University Mainz, Joh.-Joachim-Becherweg 21, 55099 Mainz, Germany
[6]Institute of Environmental Physics, University of Bremen, Otto-Hahn-Allee 1, 28334 Bremen
[7]Alfred Wegener Institute, Helmholtz Centre for Polar and Marine Research, Bussestr. 24, D-27570 Bremerhaven, Germany
[8]Leipzig Institute for Meteorology, University of Leipzig, Stephanstr. 3, 04103 Leipzig
[9]Leibniz Insititue for Tropospheric Research, Permoserstr. 15, 04318 Leipzig, Germany

**Correspondence:** Erlend M. Knudsen (eknudsen@uni-koeln.de)

**Abstract.** The two concerted field campaigns Arctic CLoud Observations Using airborne measurements during polar Day (ACLOUD) and the Physical feedbacks of Arctic planetary boundary level Sea ice, Cloud and AerosoL (PASCAL) took place near Svalbard from May 23 to June 26, 2017. They were focused on studying Arctic mixed-phase clouds and involved observations from two airplanes (ACLOUD), an icebreaker (PASCAL), as well as surface-based stations, a tethered balloon, and

satellites. Here, we present the synoptic development during the 35 day period of the campaigns, using classical near-surface and upper-air meteorological observations, as well as operational satellite and model data. Over the campaign period, short-term synoptic variability was substantial, dominating over the long-term background effect of Arctic amplification. During the first campaign week, cold and dry Arctic air from the north persisted, with a distinct but seasonally unusual cold air outbreak. Cloudy conditions with mostly low-level clouds prevailed. The subsequent two weeks were characterized by warm and moist

maritime air from the south and east, which included two warm air advections. These synoptical disturbances caused lower cloud cover fractions and higher-reaching cloud systems. In the final two weeks, adiabatically warmed westerly air dominated, with a strongly varying cloud distribution in between the two other periods. Results presented here provide synoptic information needed to analyze and interpret data of upcoming studies from ACLOUD/PASCAL, while also offering unprecedented measurements in a sparsely observed region.

*Copyright statement.* TEXT



# 1   Introduction

The phenomenon of Arctic amplification — the 2–3 times higher warming of the Arctic relative to the global atmosphere — is a major indication of current drastic Arctic climate changes (Serreze and Barry, 2011). A number of potential causes for this special feature of the Arctic climate system are discussed, which include various interconnected processes and feedback
mechanisms, such as sea ice loss and surface albedo feedback, meridional atmospheric and oceanic energy fluxes, and atmospheric radiation effects linked to temperature, water vapor and clouds. Still, the relative importance of these different feedback mechanisms is subject of the current scientific debate (Wendisch et al., 2017).

Climate models have difficulties in reproducing the observed drastic Arctic climate changes, and therefore the uncertainty in Arctic climate projections is larger than in other parts of the world (Stocker et al., 2013). This issue is related to major
gaps in understanding of key processes particularly important for the Arctic climate system. Significant uncertainties in the parameterization of subgrid-scale processes remain one of the major challenges for realistic climate simulations, particularly in high latitudes (Vihma et al., 2014). Further important open questions are associated with cloud physical processes (e.g., Tjernström et al., 2008; de Boer et al., 2014; Pithan et al., 2014) and sea ice albedo-cloud radiative interactions (e.g., Karlsson and Svensson, 2013; English et al., 2015). The results of different Arctic climate models substantially disagree; they also
generally do not match with observations in particular with respect to hydrometeor phase partitioning in mixed-phase clouds (Morrison et al., 2011; McIlhattan et al., 2017) and the vertical structure of the atmospheric boundary layer (ABL; Svensson and Lindvall, 2015), which are interrelated (Lüpkes et al., 2010; Barton et al., 2014; Pithan et al., 2014). Those biases can considerably affect the water vapor and temperature profiles and the atmospheric radiation budget, which can consequently alter the individual climate feedback (Kim et al., 2016). To make substantial progress in these areas, dedicated observational
campaigns in the Arctic are key.

In this framework, a number of airborne and ship-based campaigns with a focus on Arctic aerosol-cloud-ABL processes were conducted within the last decade (Wendisch et al., 2018, and references therein). However, most of these previous observational campaigns in the Arctic obtained relatively few process-level observations of the coupled Arctic climate system, especially related to interactions between clouds and the ABL and with regards to the radiative interaction of the cloud properties with
the surface. And, although all these campaigns have been conducted in the last decade and thus measured the "new Arctic" (Jeffries et al., 2013, and references therein), they are hard to compare due to the different synoptic and sea ice conditions as well as climate regimes in the various regions. Nevertheless, the comparison both with other campaigns and with the long-term observations of the land-based station Ny-Ålesund help to estimate the representativeness of the measurements for the sea ice environment of the Arctic North Atlantic sector, and if/how the results can be scaled up or generalized.

The Arctic CLoud Observations Using airborne measurements during polar Day (ACLOUD) and the Physical feedbacks of Arctic planetary boundary level Sea ice, Cloud and AerosoL (PASCAL; Macke and Flores, 2018) field campaigns (hereafter referred to as ACLOUD/PASCAL) were conducted from May 23 to June 26, 2017 (Wendisch et al., 2018). Concerted, process-oriented observations of a diversity of atmospheric and surface parameters were collected by instrumentation installed on the Polar 5 and Polar 6 aircraft of the Alfred Wegener Institute (AWI), an ice floe station including a tethered balloon, the research



vessel (RV) and icebreaker *Polarstern* of AWI (hereafter referred to as Polarstern), and from the ground-based site in Ny-Ålesund on Svalbard. The campaigns took place near Svalbard in the transition zone of the Greenland Sea and the Arctic Ocean between open ocean and sea ice.

The Arctic North Atlantic sector is particularly different as compared to other Arctic regions. It is frequently affected by

cyclones associated with the Icelandic low (Serreze et al., 1997), which transport heat and moisture into the Arctic, driving the transitions between radiatively clear and cloudy states (Stramler et al., 2011; Graham et al., 2017). It is also the region of most frequent intrusions of moist and warm air entering the Arctic (Woods and Caballero, 2016), which affects the marginal ice zone (MIZ) as well as the atmospheric thermodynamic structure, and the formation, distribution, and properties of clouds (Johansson et al., 2017). In this area, the conditions are favorable for studying the coupling of the ABL clouds with cyclones

and large-scale circulation. Furthermore, the proximity to the sea ice edge north of Svalbard allows an investigation of the cloud microphysical changes during air mass transformations during both moist air intrusions and cold air outbreaks (Young et al., 2016). Overall, the area close to Svalbard enables studies of the response of cloud properties to changes in local sea ice conditions, surface heat and moisture fluxes, in the thermodynamic structure of the lower atmosphere, and to the large-scale synoptical conditions that control the origin of the air mass in which the clouds form.

The intra- and interannual variability of the Arctic atmosphere is an important aspect. Therefore, it is crucial to put the short-term campaign observations into a climatological context, also to understand how representative these are. Accordingly, this paper presents the development of the synoptic conditions during ACLOUD/PASCAL and compares it with existing climatology and other Arctic field campaigns. This synoptic overview paper aims to help interpreting the upcoming detailed process studies of clouds, energy fluxes, and other parameters observed during ACLOUD/PASCAL. Moreover, our detailed

analysis gives useful insight into the processes during a typical transition period from freezing to melting conditions in the region around Svalbard. An improved understanding of processes in this region is important due to its particularly marked climate changes (Maturilli and Kayser, 2017).

Section 2 introduces ACLOUD/PASCAL and the data used to describe the synoptic conditions encountered during this period. Section 3 presents the time series of the basic meteorological variables and weather classifications. Based on these,

three key periods are defined and characterized in terms of key meteorological parameters in Sect. 4. Section 5 puts the observations into a climatological and regional context. Finally, results are summarized and concluding remarks are given in Sect. 6.

## 2   Data

In this section, we present data that were obtained during ACLOUD/PASCAL in order to characterize and classify the synoptic

evolution during the measurement period. Following an introduction of the ACLOUD/PASCAL set-up in Sect. 2.1, Sects. 2.2, 2.4 and 2.3 describe the surface-based measurements, satellites and models applied, respectively.





## 2.1 Campaign set-up

The region investigated by ACLOUD/PASCAL is shown in Fig. 1. For the comparison in Sect. 5.3, the tracks of the icebreakers *DesGroseilliers* during the Surface Heat Budget of the Arctic Ocean (SHEBA) campaign, *Oden* during the Arctic Ocean Expeditions 1996 (AOE-96) and 2001 (AOE-2001), as well as during the Arctic Summer Cloud Ocean Study (ASCOS), *Tara*
during TARA, and *Lance* during the The Norwegian young sea ICE (N-ICE2015) expedition are also included in Fig. 1a.

The set-up of ACLOUD/PASCAL is described in more detail in Wendisch et al. (2018). In addition to satellite and model data, we here also present measurement results from the land-based research station AWIPEV in Ny-Ålesund, as well as from Polarstern cruising into, mooring to and cruising out of the sea ice northwest of Svalbard.

While the climatological mean location of the MIZ runs southwest from Svalbard toward Greenland (Figs. 1b and 1c), it
extended anomalously close to Svalbard in the near west, north, and east vicinity during spring 2017 compared to recent years (Tetzlaff et al., 2014; Fetterer et al., 2018). Therefore, Polarstern was able to moor onto an ice floe relatively close to Svalbard (around 82° N, 10° E; Fig. 1c), making it easy to reach the icebreaker with Polar 5 and Polar 6 based in Longyearbyen (Fig. 1b). In addition to Ny-Ålesund, the atmospheric column over the MIZ west of Svalbard was also in reach of the aircraft and investigated during ACLOUD. Finally, to characterize the vertical structure of clouds, five flights with Polar 5 and Polar 6 were
coordinated with an A-Train satellite constellation overpass (Stephens et al., 2002).

## 2.2 Surface-based measurements

Near-surface meteorological and radiosonde data were collected throughout ACLOUD/PASCAL in Ny-Ålesund, at the ice floe station, and on-board Polarstern. The former cover the entire ACLOUD flight period May 23 – June 26, 2017, whereas the latter stem from the time when Polarstern was north of the Arctic Circle only, May 28 – June 18, 2017. These are presented in
Sects. 3.1, 3.2, and 4.1.

The AWIPEV research base in Ny-Ålesund is located about 100 km northwest of Longyearbyen. Since 1992, AWI routinely operates a variety of atmospheric measurements in Ny-Ålesund, which were intensified during ACLOUD/PASCAL. The frequency of the daily radiosonde measurements was increased to four RS41 launches per day, providing 6-hourly vertical profiles of temperature, humidity, pressure, and wind speed and direction with about 5 m vertical resolution (Maturilli, 2017b, a). The
same meteorological parameters were observed every minute at the surface (Maturilli et al., 2013), of which surface pressure and 2 m temperature are presented here.

In immediate vicinity, the surface radiation measurements of the Baseline Surface Radiation Network (BSRN) provide information on global and reflective solar radiation (Maturilli et al., 2015). Furthermore, 6-hourly integrated water vapor (IWV) is retrieved by the integration of the respective radiosonde humidity profiles, whereas daily precipitation amount is
obtained from the Norwegian Meteorological Institute (MET Norway). Additionally, specific ground-based remote sensing campaign activities regarding aerosol particles and clouds were conducted by lidar, radar, microwave radiometer, and other instrumentation, as described by Wendisch et al. (2018).




Four daily RS92 radiosondes were launched from Polarstern during most of ACLOUD/PASCAL (Schmithüsen, 2017). Together with every minute pressure observations at 16 m height and temperature observations at 29 m height on-board Polarstern (Schmithüsen, 2018), these retrieved vertical profiles are compared to the Ny-Ålesund data. For more information on the dedicated Polarstern campaign instruments, please see Macke and Flores (2018) and Wendisch et al. (2018).

## 2.3 Satellites

Polar-orbiting satellites play a key role for studies of sea ice, snow and cloud variability on a regional scale in the Arctic. Sea ice data for the ACLOUD/PASCAL region in Figs. 1b and 1c and Sect. 4.3 are obtained from the University of Bremen (UB; Spreen et al., 2017, following Spreen et al., 2008), the National Snow and Ice Data Center (NSIDC; Fetterer et al., 2018), and the Ocean and Sea Ice Satellite Application Facility (OSI SAF; Lavergne et al., 2010). They provide sea ice concentration over the ACLOUD/PASCAL measurement period, over the climatological period 1979–2017, and sea ice drift over the ACLOUD/PASCAL measurement period, respectively.

Daily sea ice concentration from UB and NSIDC were obtained at 6 km x 4 km and 25 km x 25 km resolutions, respectively. These were from the Advanced Microwave Scanning Radiometer for the Earth Observing System (AMSR-E) and 2 (AMSR2) sensors (Spreen et al., 2017), and Scanning Multichannel Microwave Radiometer (SMMR), Seasat, Special Sensor Microwave Imager (SSM/I), and Special Sensor Microwave Imager/Sounder (SSMIS) sensors (Fetterer et al., 2018), respectively. The multisensory products AMSR2 and scatterometers are combined using an advanced cross-correlation method (Continuous MCC) for the bidaily sea ice drift data, which were downloaded at a 62.5 km x 62.5 km resolution.

The spatially varying date of snow melt onset in the ACLOUD/PASCAL region relative to climatology is shown in Sect. 5.2. This analysis was based on the method by Markus et al. (2009) (and updated by J. A. Miller of the National Aeronautics and Space Administration Goddard Space Flight Center; NASA GSFC), who used the NSIDC data to develop an Arctic melt season climatology since 1979. The method utilizes the agreement of different brightness temperature criteria. Compared to other methods (e.g., buoy data), satellite passive microwave measurements have a larger spatial coverage, have a relative long and consistent record, and are directly related to the melt signature of sea ice or the overlying snow cover. The latter is largely fluctuating with snow and ice wetness, which drastically change the dielectric properties of snow and ice and therefore their emissivities.

Cloud properties are routinely retrieved from different polar-orbiting satellite instruments. Unfortunately, considering the special focus on clouds during ACLOUD/PASCAL, the most relevant satellite in the A-train constellation — CloudSat — entered standby mode June 4, 2017 (CloudSat DPC, 2017). Therefore, in Sect. 4.4, we show cloud observations made with the less advanced Infrared Atmospheric Sounding Interferometer (IASI), which is limited in vertical resolution but shows much better spatio-temporal coverage (EUMETSAT, 2017). Infrared sounders are particularly advantageous to retrieve upper-tropospheric cloud properties, with a reliable cirrus identification, day and night (Stubenrauch et al., 2017).

IASI is part of the MetOp series of polar orbiting satellites and has a swath width of about 2, 200 km (EUMETSAT, 2017). Due to the meridional convergence of the orbits the temporal sampling of the ACLOUD/PASCAL region is high, with several overpasses per day. The level 2 cloud cover fraction and top pressure products from Version 6 are used to investigate the



vertical distribution of clouds. A cloud detection is performed followed by a retrieval of cloud top pressure using the $CO_2$-slicing technique for each IASI field of view (e.g., Lavanant et al., 2011). As shown by Lavanant et al. (2011), different retrieval schemes agree best for high-level clouds and for opaque and overcast situations for the lower-level layers while multi-layer clouds pose problems. Comparisons between the schemes reveal differences of about 100–150 hPa with biases of about

50 hPa or less.

## 2.4  Models

Because in situ and satellite data can only provide a limited perspective, reanalysis and analysis data from the European Centre for Medium-Range Weather Forecast (ECMWF) are used to best describe the state of the atmosphere over the broader domain and longer time scales. Explicitly no use of forecasts are analyzed here as one of the ACLOUD/PASCAL objectives is to

investigate their skill.

The European Re-Analysis Interim (ERA-I; Dee et al., 2011) provided data of atmospheric circulation, temperature, and humidity for our region of interest. This reanalysis provides the best description of the state of the atmosphere by assimilating a wealth of observations, including satellites, radiosondes, and land stations, and is found to be well-suited for the northern regions (Jakobson et al., 2012; Chung et al., 2013; Lindsay et al., 2014).

ERA-I data were acquired on a $0.75° \times 0.75°$ horizontal grid for the period of May–June 1979–2017. In the calculation of the weather events and classifications in Sects. 3.3 and 5.2, 6-hourly 850 hPa and skin temperature and 850 hPa geopotential were used. Parameters presented in Sect. 4.2 are based on 6-hourly 700 hPa geopotential, zonal and meridional winds, temperature, and specific humidity. Daily 1000 hPa geopotential was obtained for Sect. 5.1.

ECMWF analysis data were obtained on a $0.25° \times 0.25°$ horizontal grid. These data provided input into the Lagrangian

particle dispersion model Flexible Particle Dispersion (FLEXPART; Stohl et al., 2005) used to analyse the history of air masses arriving in Ny-Ålesund in Sect. 4.1.

## 3  Temporal evolution

In this section, we analyze time series from Ny-Ålesund and Polarstern over the course of the ACLOUD/PASCAL measurement period. These are meteorological parameters from near-surface and radiosonde observations in Sects. 3.1 and 3.2, respectively.

Time series of weather classifications based on reanalysis data follows in Sect. 3.3. A more detailed description of the day-to-day weather development as observed by Polarstern can be found in Macke and Flores (2018).

### 3.1  Near-surface meteorological observations

Figure 2 shows time series of key meteorological parameters from Ny-Ålesund and Polarstern during the ACLOUD flight period and PASCAL ocean-cruising and ice-attached period, respectively. In the former, the permanent AWIPEV and MET

Norway weather stations allow a comparison to the observed long-term average 1993–2016. For the detailed description of key synoptic days, we also provide weather charts from www.wetterzentrale.de in Fig. A1 in the appendix.



As observed in Ny-Ålesund, the ACLOUD flight period started in a cold and dry period under low surface pressure influence with northerly winds (Fig. 2). This pattern is indicative of a marine cold air outbreak (MCAO; Kolstad, 2017).

After about three days, the pattern started to change, which finally led to the onset of the melt period in Ny-Ålesund on May 29. The first indication of a change was a pressure increase in Ny-Ålesund and variable wind direction over Svalbard with one
episode (on May 26) of southerly wind in Ny-Ålesund (Fig. 2a). The variable winds were caused by a low pressure system approaching Svalbard from the northeast (Fig. A1a). However, over the Fram Strait, strong northwesterly flow with convective conditions prevailed until May 26.

The variable wind direction over Svalbard finally caused the highest precipitation observed in Ny-Ålesund during ACLOUD/PASCAL, with 2 mm observed on May 27 (Fig. 2c). A substantial increase in IWV (from 6 kg m$^{-2}$ on May 28 to 14 kg m$^{-2}$ on May
30; Fig. 2c) and temperature (from $-10°$C on May 29 to $+7°$C on May 31; Fig. 2b) followed, resulting from easterly winds during a warm front passing over Ny-Ålesund. This was part of a low pressure system moving westward south of Svalbard (Fig. A1b).

A few days later, strong southwesterly flow developed west of Svalbard. This caused a melt onset also over the northern Fram Strait. The dramatic change of wind was again caused by a high pressure ridge that diagonally extended over Svalbard,
with a center southwest of the archipelago (Fig. A1c). In Ny-Ålesund, this explains the increasing surface pressure up to 1029 hPa on June 2 (Fig. 2a). Coincidentally, a low pressure system developed over northern Greenland. These two pressure systems north and south of Svalbard led to strong southwesterly air advection across the northern Fram Strait, with an atmospheric river reaching the ACLOUD/PASCAL region from the east (not shown). On the northerly cruising Polarstern in the waters west of Spitsbergen (Fig. 1c), temperatures rose from $-2°$C on May 29 to $+7°$C on May 31 (Fig. 2b). The peculiarly warm
temperatures from May 28 until late May 29 presumably resulted from the southerly location of Polarstern at this point (near 70°N), as also indicated by the apparent mismatch in near-surface pressure with Ny-Ålesund (Fig. 2a). Similarly, the rapid cooling observed on Polarstern from $+7°$C to $-6°$C over May 31 coincided with its entrance into the sea ice northwest of Spitsbergen (Fig. 1c).

The 17°C warming over only two days in Ny-Ålesund, which marked the beginning of the snow melt season on May 29
(Fig. 2b), was also recognized by surface radiation measurements (i.e., albedo). From this date, the surface albedo dropped from almost 0.9 to below 0.1 by June 14, when the snow had completely disappeared. In comparison, the first snow-free day has been detected between May 30 and July 5 since the beginning of the BSRN measurements in Ny-Ålesund in late 1992.

It should be noted that due to the southerly winds over the Fram Strait during this time frame, the highest temperatures relative to the long-term observations were recorded during ACLOUD/PASCAL as the wind direction again turned southerly.
In Ny-Ålesund, 7°C and 8°C were observed on May 31 and June 6, respectively (Fig. 2b), both being indications of warm air advections (WAAs; Tjernström et al., 2015). The latter event was accompanied by a relative IWV peak of 15 kg m$^{-2}$ (Fig. 2c), thus indicating a moist air intrusion (Woods and Caballero, 2016).

June 6 was also the date when the observations from the ice-attached Polarstern started. Over its first days in the ice, Polarstern observed an increase in near-surface pressure due to a high pressure ridge east of Svalbard (Fig. A1d), peaking at
1029 hPa on June 8, and associated with rises in IWV from 6 kg m$^{-2}$ to 17 kg m$^{-2}$ on June 9 (Fig. 2c) and in near-surface





air temperature from $-8°$ C to $+2°$ C on June 10 (Fig. 2b). In other words, the above-freezing temperature on Polarstern while surrounded by sea ice (June 1–17) occured four days after that in Ny-Ålesund. This delay could be explained by its more northerly location in the ice, where a temperature inversion with surface cooling during warm air advection across the MIZ is likely. Hence, as long as the inversion is not destroyed, it remains cold at the lowest levels. Anomalously warm and moist

air was also observed in Ny-Ålesund these days, but with less intense changes due to the already warm and moist air since June 6. Thus, while the synoptic conditions were similar for Ny-Ålesund and Polarstern during June 6–8 (Fig. 2), local factors (e.g., sea ice distribution) probably played an important role for the difference between the two stations at about 335 km apart. Moreover, of the three variables compared in Fig. 2, temperature is obviously most affected by surface-based inversions (shown later in Fig. 4a).

Both Ny-Ålesund and Polarstern experienced distinct drops in near-surface pressure associated with relative peaks in near-surface air temperature and IWV around June 13 (Figs. 2a to 2c). The air masses then originated in the European source region, but curled around Svalbard and arrived Ny-Ålesund from the north (shown later in Fig. 6c). For the remainder of the measurement period, surface pressure, near-surface air temperature, and IWV observed in Ny-Ålesund were close to the long-term average, as well as close to Polarstern values until the icebreaker left the ice (June 18).

With the exceptions described above, Ny-Ålesund and Polarstern observations in Fig. 2 are comparable, meaning that they mostly were under the same synoptic influences. Comparable observations between Ny-Ålesund and their research vessel north of Svalbard were also found during the N-ICE2015 expedition (Kayser et al., 2017). It is, therefore, appropriate to set the observations during the ACLOUD/PASCAL measurement period into context with the long-term observational record from Ny-Ålesund.

**3.2  Radiosonde observations**

To further investigate how the surface, the boundary layer, and the free troposphere are coupled, time series of temperature and specific humidity profiles over the ACLOUD/PASCAL measurement period from Ny-Ålesund and Polarstern are shown in Figs. 3 and 4. Specific humidity in Figs. 3b, 3d, and 4b was calculated using the vapor pressure formulations by Hyland and Wexler (1983). ABL heights in Figs. 3 and 4 are identified using the surface-based bulk Richardson number approach assuming

a critical value $Ri = 0.25$, as discussed in Hanna (1969), Zhang et al. (2014), and Kayser et al. (2017). These heights will be discussed in association with Fig. 4.

For readability, time series of wind profiles are included in Figs. 3c and 3d from Polarstern only. However, the time series of 850 hPa wind from Ny-Ålesund is shown in Fig. 2a.

The daily long-term radiosonde records, following Maturilli and Kayser (2017), demonstrate the increase in air temperature

and specific humidity from May 23 to June 26 (Figs. 3a and 3b). By the beginning of June, near-surface air temperature (specific humidity) usually exceeds $0°$ C ($3 \, \mathrm{g \, kg^{-1}}$).

As indicated by the near-surface air temperature observations in Ny-Ålesund (Fig. 2b), the anomalously cold first week with subfreezing temperatures was followed by two exceptionally warm weeks partially above $5°$ C (Figs. 3a and 3c). This rapid change around May 30 obviously occurred throughout the entire tropospheric column.





The WAA starting around May 29 over Ny-Ålesund only shortly enhanced tropospheric humidity levels (Fig. 3b). No significant changes in specific humidity were associated with this event over Polarstern, where values around 3 g kg$^{-1}$ during the period of May 31 – June 5 were limited to the lowest 100 hPa (Fig. 3d). This situation changed during a second WAA and first sustaining moist air intrusion on June 6, when the temperature in the lowest 300 hPa over Ny-Ålesund (Polarstern) reached up to $+8°$ C ($+4°$ C) and humidity 5 g kg$^{-1}$ (4 g kg$^{-1}$) over about a week, exceeding long-term averages with a peak around June 8–12.

The vectors in Figs. 3c and 3d indicate that the highest temperatures and humidities occurred in association with a shift to generally easterly winds below 700 hPa (southerly below 850 hPa during June 7–9). Air from the east (south) warmed and moistened over the open ocean west and southwest of Franz Josef Land (Spitsbergen). Above 700 hPa, a northerly wind component dominated.

However, the prevailing winds changed during June 11 and 12. Then northerly winds dominated the lower troposphere, indicating the end of the moist air intrusion. For the remainder of the measurement period, the temperature and specific humidity over Ny-Ålesund remained around the long-term averages.

Figure 4 gives a closer look at the lowest 2000 m over Polarstern. On a background of temperature and specific humidity from Fig. 3c and 3d, respectively, ABL heights are plotted along with the extents of surface-based and lifted inversions. For each sounding, the lowermost inversion — if there is one — is shown, which is either connected to the surface or is lifted.

Following the work by Kahl (1990), Andreas et al. (2000), and Kayser et al. (2017), an inversion is identified when vertical temperature (humidity) gradients of at least 0.3 K (0.3 g kg$^{-1}$) per 2 m were detected. Resulting inversion layers with a vertical extent less than 100 m were neglected. The extent of the inversions is quite sensitive to the choice of this objectively chosen value, which was selected to avoid resolving too small features. In general, humidity inversions should be interpreted with care, as humidity is distributed much more inhomogeneously than temperature.

Before May 30, the ABL was relatively high (up to 1300 m) and only few inversions were observed, probably due to the relatively southerly location of Polarstern in the open Arctic Ocean (Fig. 1c). After this date, the ABL height generally varied between 100 and 800 m, with inversions almost continuously observed (Fig. 4). When the ABL was relatively high (up to 700 m), a lifted temperature inversion was present, while a surface-based temperature inversion was observed when the ABL was shallow (about 200 m).

This pattern was particularly evident when Polarstern was fixed to the ice floe (June 6–16). Then, the temperature profile was modified by both surface energy fluxes and advective features, which is consistent with the results reported by Kayser et al. (2017) for the N-ICE2015 campaign. This was exemplified during the moist air intrusion around June 10. The ABL height then increased, while the temperature inversion switched from surface-based to lifted. When the surface cooled around June 11, the ABL height decreased and a surface-based temperature inversion was re-established.

### 3.3 Weather classification

In Fig. 5, the weather during ACLOUD/PASCAL is put into the context of synoptic atmospheric patterns by analyzing the temporal evolution of an index for MCAOs and by classifying the weather situations in the campaign region. These offer better





understanding of the local weather as compared to the large-scale Arctic Oscillation and Dipole indices (shown for comparison in Fig. A2 in the appendix and for large-scale reference later in Fig. 11).

Following Papritz et al. (2015) and Kolstad (2017), the MCAO index is defined as difference between surface and 850 hPa potential temperature of each grid point, area-averaged over the eastern Greenland Sea (here defined 75.00–80.25° N, 4.50–10.50° E). Land grid cells and cells for which the surface temperature is lower than 271.5 K are not included in the area-averaging. Time series of the 6-hourly MCAO index are used to identify events of cold air outbreaks. A new event begins when the index is greater than 0 K and ends if the index falls below 0 K. Then, the last time for which the MCAO index > 0 K is set as the final time step of the event. Events are recorded only if an index value of at least 2 K is reached and the duration is at least 48 hours. The peak of each event is required to occur within the ACLOUD/PASCAL measurement period, but the events are allowed to start any time in May or by the end of June. The threshold of 2 K is defined lower than in studies focusing on the cold season (e.g., 3 K in Kolstad, 2017). This is to account for the fact that MCAOs occur considerably less frequent and are considerably less severe in late spring than in winter (Fletcher et al., 2016).

The time series also indicate WAAs. While MCAOs are characterized by a change of atmospheric stratification toward stable conditions, i.e., positive values of the MCAO index, WAAs are identified by a strongly negative deviation of the MCAO index relative to the long-term average. Here, we use a threshold of −10 K in the difference between the actual MCAO index and the average over 1979–2016 to initialize a new WAA event, before the procedure follows that for the MCAO events. These events will be discussed more in detail in Sect. 5.2.

Over the first three weeks of ACLOUD/PASCAL, the MCAO index varied considerably (Fig. 5a). During the first eight days (May 23–30), values were above the median and mostly exceeded the 95 % percentile until May 28. Corresponding to the anomalously cold and dry air observed in Figs. 2 and 3, we identify a MCAO event during the first week of the measurement period (maximum May 23 in Fig. 5a). The MCAO index then dropped significantly from +2 K on May 28 to −11 K on May 31, remaining below the median until June 15. During these two weeks, values remained below −12 K (i.e., below the 25 % percentile) except for June 7. Hence, referring again to Figs. 2 and 3, we identify two WAA events during the second and third week of the measurement period (minima June 5 and 10 in Fig. 5a). After June 14, the MCAO index increased again and leveled around the long-term median between −5 K and −7 K, indicating normal weakly unstable conditions in the lower troposphere (i.e., neither MCAO nor WAA conditions).

Adding to the MCAO index, Fig. 5b shows the Jenkinson-Collison circulation weather type (CWT) classification (Jenkinson and Collison, 1977) for Ny-Ålesund (here defined 77.25–81.00° N, 9.75–14.25° E) and Polarstern (here defined 80.25–84.00° N, 7.50–12.75° E) over the ACLOUD flight and PASCAL ice-attached periods, respectively. These are computed with the FORTRAN software COST 733 (Philipp et al., 2016) using the 6-hourly 850 hPa geopotential from ERA-I.

In accordance with the wind pattern in Fig. 2a and positive MCAO index in Fig. 5a, a northerly/northwesterly CWT dominates over Ny-Ålesund during the first five days of ACLOUD/PASCAL. However, four days of northeasterly CWT follow the cyclonic CWT on May 28, representing the southward-moving low pressure system over Ny-Ålesund, as discussed in Sect. 3.1.





A northeasterly wind direction over Ny-Ålesund means air advected across Spitsbergen, with potential for adiabatic heating down the glacier Isachsenfonna and thermodynamic heating and moisture uptake over the fjord Kongsfjorden northeast and east of Ny-Ålesund, thus in line with the temperature and humidity increase in Figs. 2b, 2c, 3a, and 3b. The dominating anticyclonic and westerly CWT during June 2–13 over Ny-Ålesund and Polarstern both transport air advected over the open ocean, thus favoring even higher temperatures and humidity.

This air regime broke off on June 14, as the CWT remained northerly over Ny-Ålesund and Polarstern for about three days. For the remaining 1.5 weeks of ACLOUD/PASCAL, CWTs varied considerably. Hence, no distinct air advection was observed and only small changes were measured in temperature and specific humidity (Figs. 2b, 2c, 3a, and 3b).

## 4 Key period characteristics

In this section, we highlight the characteristics of three key periods, as defined based on the time series shown in Sect. 3. These serve as the basis for the regional and local meteorological data shown for each of the key periods in Sects. 4.1, 4.2, 4.3, and 4.4.

Based on Figs. 2-5 (cf. discussion in Sects. 3.1 to 3.3), we define the following key periods during the ACLOUD/PASCAL measurement period:

1. The cold period (CP) — May 23–29, 2017 (7 days)

2. The warm period (WP) — May 30 – June 12, 2017 (14 days)

3. The normal period (NP) — June 13–26, 2017 (14 days)

The three key periods represent three different synoptic tendencies and not states. For example, CP ends as the near-surface air temperature in Ny-Ålesund starts rising (on May 29) and not when it exceeds 1 standard deviation (on May 31) (Figs. 2b and 3a).

### 4.1 Air mass distribution

FLEXPART was used in backward mode to analyse the air mass history during the three key periods defined above. For this, we continuously released particles within FLEXPART close to the surface at the location of Ny-Ålesund over all 24 hours every day during the ACLOUD/PASCAL measurement period. The particles represent an inert, non-interacting tracer and are traced back in time for another ten days.

In Fig. 6, we show the potential emission sensitivity (PES; Stohl et al., 2005), which is related to the time an air parcel spends in a particular region over a particular time period before arriving in Ny-Ålesund. We further integrate PES in the vertical and show values for the total tropospheric column along with center of mass trajectories for each day in the three key periods. To distinguish between the three key periods, we focus our analysis on temporal means of PES over the corresponding periods.



During CP, most air masses were located within $70°$ N without significant midlatitude influence, as indicated by Fig. 6a. Their origin were mostly the central and eastern parts of the Arctic Ocean, with smaller contributions from the Siberian coast, the Canadian Arctic, and Greenland. This Arctic air was cold and dry, as indicated in Figs. 2b, 2c, 3a, and 3b.

In the twice as long WP, the trajectories spanned a larger area, originating as far south as $50°$ N (Fig. 6b). Three major areas
then influenced the air mass characteristics reaching Ny-Ålesund: northern Europe, Siberia, and the Arctic North Pacific sector. There was only a weak influence from the North American archipelago and the central Arctic Ocean.

During the first WP week, the air parcels entered the Nordic Seas from the eastern Arctic Ocean, either crossing over Svalbard from the east or going around the archipelago to arrive in Ny-Ålesund from the southwest. These two pathways are expected to warm the air masses adiabatically across Spitsbergen or thermodynamically over the ocean, respectively (cf. discussion in
Sect. 3.3). We identify these two patterns in Figs. 3a and 3b, where the latter pattern is also characterized by higher humidity from the ocean. While originating in the Nordic Seas or over northwestern Eurasia, air masses during the second WP week also crossed open water before reaching Ny-Ålesund from the south, thus being similar to the latter of the two described patterns.

The pattern for the last key period — NP — was a mixture of the two former key periods. Most of the Arctic Ocean and the Nordic Seas were then sources of air mass origin, but the highest density was found in air arriving Ny-Ålesund from the west
(Fig. 6c). Whether passing over the sea ice north of Greenland, the open ocean south of Svalbard or the Greenland ice sheet and thus susceptible to adiabatic heating, the air masses reached Ny-Ålesund from the sea ice/open ocean transition zone in the Fram Strait. Overall, this led to relatively average temperate and humid air as observed in Ny-Ålesund (Figs. 2b, 2c, 3a, and 3b).

Figure 7 shows the varying profiles of temperature and specific humidity as observed over Ny-Ålesund and Polarstern during
the three key periods. Only Ny-Ålesund data are included in Figs. 7a and 7b due to the southerly location of Polarstern during the first campaign week, unrepresentative of the Arctic. In Figs. 7c to 7f, Polarstern data are split in two profiles to differentiate its ice-attached and ocean-cruising locations (cf. Fig. 1c).

While in the end of May, the first key period (CP) was characterized by relatively cold and dry air above Ny-Ålesund, with temperatures continuously below $0°$ C and humidity mostly below 2 g kg$^{-1}$ (Figs. 7a and 7b). The nearly isothermal average
profile between 900 and 800 hPa is consistent with the top of the frequent low-level clouds observed during this period (shown later in Fig. 10b). Low variability was then found in the specific humidity profiles, although some soundings revealed inversions above 800 hPa. This was probably an effect of the radiosondes escaping the mountains surrounding Kongsfjorden and entering the free troposphere.

No distinct temperature inversions were observed over Ny-Ålesund during the second key period (WP) either, but the air
below 500 hPa warmed substantially, with temperatures above the freezing point at the surface for all the soundings (Fig. 7c). Compared to CP, temperatures during WP were typically about $10°$ C warmer in this column (Fig. 7c compared to Fig. 7a). Also humidity levels below 500 hPa were remarkably higher, with specific humidity values up to 5 g kg$^{-1}$ (Fig. 7d compared to Fig. 7b).

The mean temperature profiles of the ocean-crossing and ice-attached Polarstern were similar to those of Ny-Ålesund above
900 hPa during WP (Fig. 7c). However, in the lowest 100 hPa, a mean temperature inversion with 5 K inversion strength




(slightly less for the ocean-crossing profiles) was found. Moreover, the variance of the ocean-crossing Polarstern profiles differed, with lower minimum values.

For specific humidity, the variance over Ny-Ålesund overlapped with both the ocean-crossing (lower range) and ice-attached profiles (upper range) from Polarstern during WP (Fig. 7d). Both Ny-Ålesund and the ice-attached Polarstern observed some
strong humidity inversions between 900 and 800 hPa, as explained by the moist air advection during the second week of this key period (Figs. 3b and 3d). In the preceding week, humidity inversions were found between 800 and 750 hPa over the ocean-crossing Polarstern.

Average values in the lowest 300 hPa were typically about $1\,\mathrm{g\,kg^{-1}}$ higher over Ny-Ålesund than over the ocean-crossing Polarstern during WP (Fig. 7d), highlighting the difference between the ice-free Kongsfjorden near the former and the patchy
ice around the latter at this point. However, while in the ice, the average humidity profile from Polarstern had lower values at the surface ($3\,\mathrm{g\,kg^{-1}}$) and higher values in the lower 300 hPa (typically about $1\,\mathrm{g\,kg^{-1}}$ higher than Ny-Ålesund) than the other two profiles.

The Ny-Ålesund average temperature profile during the last key period (NP) had a similar shape as during CP (Fig. 7e compared to Fig. 7a). However, it was about a $10°\mathrm{C}$ warmer throughout the atmospheric column, and both the near-isothermal
behavior and largest variance were found at about 25 hPa lower heights. These features were similar in the ice-attached and ocean-cruising Polarstern profiles, although here inversions were present around 925–900 hPa (Fig. 7e).

Inversions were also observed in the Polarstern average specific humidity profiles during NP around 925 hPa (Fig. 7f; the lowermost few meters inversion-like features at the surface are likely an artifact due to the start of the balloon launch process or temporal inhomogeneous sampling during pressure conditions of more than 1006 hPa). Compared to CP, both average values
and variances were higher (Fig. 7f compared to Fig. 7b). It should be noted, however, that the ice-attached and ocean-cruising profiles during NP in Figs. 7e and 7f only are based on four and two days, respectively.

## 4.2 Atmospheric circulation and thermodynamics

The contrasting atmospheric circulation, temperature, and humidity of the three key periods are analyzed in Fig. 8. Here, Figs. 8a, 8c, and 8e illustrate the 700 hPa geopotential height and horizontal wind of CP, WP, and NP, respectively, while the
relative temperature and humidity of these periods are depicted in Figs. 8b, 8d, and 8f. For a more detailed evolution during ACLOUD/PASCAL, daily fields of these atmospheric circulation and thermodynamic measures are presented in Figs. A3 and A4 in the appendix. The 700 hPa virtual potential temperature is a merged measure of air temperature and humidity, and is estimated from the 700 hPa temperature and specific humidity fields (Etling, 2008). The choice of 700 hPa follows from the main flight level during ACLOUD (Wendisch et al., 2018).
Figures 8a and 8b extends the pattern shown for the first campaign week (CP) in Figs. 2, 3, and 6. A northerly air flow west and north of Spitsbergen at 700 hPa follows from the anomalous low geopotential height centered over the Pechora Sea (Fig. 8a). The dry and cold Arctic air brought virtual potential temperatures down to $-9°\mathrm{C}$ lower than climatology in the Barents Sea (Fig. 8b). Similarly, temperatures were $4–8°\mathrm{C}$ below the climatology of $13–17°\mathrm{C}$ in the ACLOUD/PASCAL region.



There was a marked change in atmospheric circulation during the next two weeks of ACLOUD/PASCAL (WP; Fig. 8c compared to Fig. 8a). While the climatology did not change much, the anomalous high 700 hPa geopotential height centered over the Fram Strait, Svalbard, and north of the archipelago caused an anticyclonic wind pattern in the region (Fig. 8c). Moist and warm maritime air were advected from the Norwegian and Greenland seas into the region, with virtual potential

temperature values reaching $8°$ C above the climatology at the ice edge northwest of Spitsbergen (Fig. 8d). Relative to Cap of the North, which then were in a northeasterly wind regime (Fig. 8c), the ACLOUD/PASCAL region was about $5°$ C warmer and moister (Fig. 8d).

During the final two weeks of ACLOUD/PASCAL (NP), 700 hPa atmospheric circulation resembled that of the first week, but with no distinct minimum in geopotential height anomalies (Fig. 8e compared to Figs. 8a and 8c). Instead, the lowest values

were generally found from Novaya Zemlya to Franz Josef Land. This meridional anomaly contrasted the climatological trough over the Greenland Sea and caused a northwesterly air flow around Svalbard (Fig. 8e). As a result, 700 hPa virtual potential temperature values were close to its climatology, generally in the range $0–2°$ C west of $15°$ E (including the ACLOUD/PASCAL region) and $−2–0°$ C east of this meridian (Fig. 8f). While the air came from the Arctic, its northwesterly origin in NP compared to northeasterly in CP allowed adiabatic heating over the Greenland ice sheet (cf. discussion in Sect. 4.1). Furthermore, during

NP, the sea ice melted substantially northeast of Greenland (shown later in Fig. 9c). Hence, the relative warm and moist water underneath likely altered the Arctic air above.

## 4.3 Sea ice dynamics

To answer the question whether the characteristic key periods also were detectable in sea ice dynamics, the sea ice concentration, edge, and drift are investigated in Fig. 9. Common for all three periods, the position of the sea ice edge did not change

much in the Fram Strait. Sea ice concentration was anomalously high in the MIZ west (typically 20–30 %), north (typically 40–50 %), and east (typically 30–40 %), respectively, while anomalously low south (typically 30–40 %) of Svalbard (cf. discussion in Sect. 2.1). Even so, there were marked changes in sea ice dynamics throughout ACLOUD/PASCAL.

During CP, the northerly wind (Fig. 8a) caused a strong southerly to southwesterly sea ice drift of about 10 km day$^{−1}$ and a positive concentration anomaly in the Fram Strait (Fig. 9a). This was particularly pronounced north of Svalbard, with values

up to 50 % above climatology.

The southerly wind during WP (Fig. 8c) reduced the sea ice drift out of the Fram Strait (Fig. 9b). Instead, the sea ice compacted, resulting in the narrower band of anomalous high sea ice concentration (5–30 % above climatology) near the ice edge north and west of Svalbard.

The band of anomalous high sea ice concentration did not change much during NP (Fig. 9c). Then, the northwesterly wind

(Fig. 8e) enhanced the ice export into the Barents Sea and contributed to the formation of the Northeast Water Polynya (Fig. 9c). The polynya, described by Pedersen et al. (1993; as cited in Schneider and Budéus, 1997), is a common phenomenon, but opened faster than usual and extended further north this year, as indicated by down to 30 % lower sea ice concentration compared to the climatology off the Greenlandic peninsula Kronprins Christian Land.





## 4.4 Cloud distribution

With ACLOUD/PASCAL aiming at investigating the role of clouds in the Arctic climate system, the question whether clouds also show a characteristic behaviour in the three key periods becomes immanent. To answer this question, we compare the average cloud cover fraction over the Nordic Seas and the central ACLOUD/PASCAL region for each key period in Fig. 10 left panels. This cloud distribution investigation is extended with an analysis of cloud top pressure in the two regions for each key period in Fig. 10 right panels. Additionally, Fig. A5 in the appendix shows time series of the daily cloud cover fraction and top pressure over the ACLOUD/PASCAL measurement period.

The cloud top pressure provides information about the vertical location of clouds. It is important to note that the passive sensors used to derive this product (cf. Sec. 2.3) can only provide information from the uppermost opaque cloud level, meaning that high-level clouds can mask low-level clouds when both layers are present. High cloud top pressure values indicate lower-level clouds, while low values are related to either upper-level clouds or clouds of larger vertical extent, which in the Arctic often are associated with synoptic systems.

Of the three key periods, the highest cloud cover fraction is observed during CP, with an average of about 85 % in the central ACLOUD/PASCAL region (Fig. 10a). In general, the highest cloud cover is observed over the open ocean (cf. Fig. 9a). This is in agreement with the results by Chan and Comiso (2013), who found a cloud cover fraction of about 88 % over ocean across the whole Arctic and all seasons.

Clouds during CP were dominated by low-level clouds with a mean top pressure of about 800 hPa (corresponding to about 1.5 km; Fig. 10b), typical for the MCAO discussed in Sects. 3.1 to 3.3. This cloud regime is also well in alignment with the reduced 700 hPa geopotential height and virtual potential temperature in Figs. 8a and 8b, indicating that the region was dominated by a northerly flow (cf. Fig. 5b). Subsequently, low-level clouds developed over the open ocean and the cloud top longwave cooling led to a temperature inversion above the cloud (cf. Fig. 7a).

According to the time series of daily cloud cover fraction and top pressure (Fig. A5), the first six days of CP can clearly be classified as a stratus regime, which Eastman and Warren (2010) found to account for the majority of Arctic clouds in the May and June climatology. On the seventh and last day of CP, the change into another circulation regime is seen as the occurrence of high level cloud (up to 300 hPa) increases in the central ACLOUD/PASCAL region, although this change is not observed near the surface (Fig. 2b).

During WP, the lowest cloud cover fraction during ACLOUD/PASCAL was observed, with an average of about 65 % and a high spread between the 15 and 95 percentile (Fig. 10c). Also the individual days were characterized by a high spread in cloud cover fraction (Fig. A5a). While CP shows a meridional band with high values of cloud cover fraction associated with the location of open ocean, the spatial distribution changed strongly in WP, with the lowest cloud cover extending from the Fram Strait northward (Fig. 10c).

The lower cloud cover fraction during WP is associated with a change in cloud type, as cloud top pressure values were around 150 hPa lower than in CP (Fig. 10d compared to Fig. 10b), highlighting the highest clouds observed during ACLOUD/PASCAL. A value of 650 hPa is typical for mid-level clouds, but can also result from a mixture of high- and low-level clouds. Average





cloud top pressure values were also more homogeneous over the Nordic Seas in WP compared to CP. Clouds were then likely associated with synoptic disturbances, which brought moister air masses from both westerly and easterly directions (cf. Fig. 5b).

The cloud cover fraction in NP was in between those of CP and WP, with an average of about 80 % (Fig. 10e). On the other hand, cloud top pressure values were around 650 hPa in the central ACLOUD/PASCAL region as in WP, but with larger spread (Fig. 10f). The strong variability was also observed on a day-to-day basis (Fig. A5), which was caused by a mix of low-, mid-, and high-level clouds. During this period, the air flow was dominantly northwesterly, and the proportion of low-level clouds increased with respect to WP.

Overall, the observed cloud cover fraction between 70 and 80 % during ACLOUD/PASCAL is in agreement with previous studies in the Arctic (Eastman and Warren, 2010; Chan and Comiso, 2013). Specifically for the Svalbard region, Mioche et al. (2015) found an average cloudiness of about 80 % for May and June using the most accurate vertical profiling satellite instruments. However, the analysis of the ACLOUD/PASCAL measurement period revealed that cloud characteristics show strong variability in space and time differing from the climatological distribution, with enhanced cloudiness over the open ocean southwest of Svalbard, while cloud cover fraction over ice-covered areas was found to be lower for most of the time. The highest contrast of cloudiness over these different surfaces is observed during CP, when the MCAO continuously triggered the formation of clouds over the warm open water. Mioche et al. (2015) identified these clouds predominantly as mixed-phase clouds (up to 60 % of all clouds). As passive satellite sensors have difficulties identifying cloud phase and multi-layer clouds, this will be investigated in more detail using other ACLOUD/PASCAL observations.

## 5 Climatological context

In this section, we present the ACLOUD/PASCAL synoptic data in regional and climatological contexts in Sects. 5.1 and 5.2, respectively. Additionally, the data are compared to other relevant Arctic field campaigns in Sect. 5.3.

### 5.1 Large-scale dynamics

The large-scale atmospheric circulation patterns are analyzed in terms of the indices Arctic Oscillation (AO; Thompson and Wallace, 1998) and Arctic Dipole (AD; Wu et al., 2006; Wang et al., 2009) over the ACLOUD/PASCAL measurement and key periods, as well as corresponding periods over 1998–2016 in Fig. 11. While the base period used in the analysis extends back to 1979, the AO and AD indices are presented for the last 20 years when the Arctic amplification has become more prominent (Serreze and Barry, 2011). This allows a more relevant comparison to the 2017 ACLOUD/PASCAL campaign.

The AO and AD indices represent the first and second leading empirical orthogonal function (EOF) modes of the daily 1000 hPa geopotential height anomalies poleward of $20°\,N$ and $70°\,N$, respectively, normalized by the standard deviation of the monthly index. Another important circulation pattern in the Northern Hemisphere is the North Atlantic Oscillation (NAO), which is characterized by a pronounced north-south dipole in sea level pressure across the North Atlantic. The NAO is in this respect very similar to the AO but without the centers of action — the Aleutian Low and the Pacific High — over the





Pacific Ocean. Accordingly, AO and NAO are closely related, with NAO actually being considered the regional occurrence of the hemisphere-wide pattern of AO (Thompson and Wallace, 1998). The analysis therefore focused on AO to provide broader information on the large-scale dynamics.

AO and AD are measures of the zonal and meridional wind patterns. AO describes the variability in the strength of the
polar vortex. A positive AO index is associated with a lower-than-average pressure over the Arctic, a strong polar vortex, and a mainly zonal jet structure. Cold polar air mass is therefore more confined and located further poleward. In contrast, a negative AO index is linked to higher-than-average pressure over the Arctic, a weaker vortex, and a stronger meridional component of the jet stream. As a result, positive AO indices correlate with more numerous and deeper cyclones in the Arctic region, while negative indices are associated with colder and wetter conditions in midlatitudes and with MCAO events in high-latitudes
(Simmonds et al., 2008; Overland et al., 2015).

A positive AD index is associated with a positive surface pressure anomaly over the Beaufort Sea, the Canadian Arctic archipelago, and Greenland, as well as a negative surface pressure anomaly over the Kara and Laptev seas. It is related to enhanced geostrophic wind flow from the Bering Strait toward the North Pole and across the Fram Strait, causing sea ice export out of the Arctic basin via the Fram Strait and the northern Barents Sea. A negative AD index is related to a lower-than-
average surface pressure over the Beaufort Sea and Greenland, a higher-than-average surface pressure over northeast Eurasia, and enhanced poleward geostrophic wind flow (Wang et al., 2009). Since the 2000s, AO is less correlated with Arctic sea ice variability than AD. A positive AD is considered the main driver of Arctic sea ice export, regardless of the sign of AO (Thompson and Wallace, 2001; Wang et al., 2009; Overland et al., 2012; Smedsrud et al., 2017).

Over the comparison period from 1998 to 2017, the AO index varied between $-1.8$ and $1.9$, with a relatively regular year-to-
year alternation between positive and negative phases until 2006 (Fig. 11a). After 2006, the fraction of positive AO index values decreased from about a half to about a third (36 % and 27 %, respectively) for the periods May 30 – June 12 and June 13–26, corresponding to WP and NP, respectively. The fraction of positive AO index values in the period May 23–29, corresponding to CP, remained at about 55 %. Nevertheless, negative AO index values dominated overall from 2007, with minima down to $-1.8$ in the three consecutive years 2010 to 2012 and in 2016. This shift toward a more dominant negative phase of the AO pattern
has been already reported in a range of recent studies and is supposed to result from the Arctic amplification of global warming (Overland et al., 2015, and references therein). In 2017, barely positive and moderate negative AO indices were found during CP and WP, respectively, which can be interpreted as an indication for enhanced meridional air mass transfer during the MCAO and WAAs (cf. Sect. 3.3). A strongly positive value of $0.9$ occurred in NP, which is in agreement with the cyclonic-dominated synoptic conditions over the Svalbard area (Fig. 5b).
AD index values ranged from $-2.2$ to $2.3$ over the comparison period 1998–2017, with the largest positive occuring in the corresponding NP of 2000 and negative values in the corresponding CP of 2002 (Fig. 11b). Since 2007, the AD index has mainly been positive, with strongly positive values from 1 to 2 dominating all corresponding key periods. The number of occurrence and magnitude of positive AD indices have increased in these years, especially in the corresponding WP by about 40 %. Only positive AD index values occurred in the years 2007, 2010, and 2012. This matches the observation that there has been
an increase in sea ice export through the Fram Strait of approximately 6 % per decade since 1979, with the highest trend of 11



% in spring and summer (Smedsrud et al., 2017). All years with strongly positive AD indices in Fig. 11b were among the years with record low sea ice extent during the last decade. Nonetheless, the number of strongly negative AD index values (below −0.8) has increased by more than 30 % over the last 11 years, and since 2013, more negative AD indices have returned. The results, therefore, point toward a general enhancement of poleward and equatorward air flow in early summer in recent years. In 2017, the AD index was positive in CP and strongly positive in NP with a value of 1.3. A slightly negative index is found for WP. The evolution corresponds to and explains well the sea ice conditions in the Fram Strait during ACLOUD/PASCAL, as described in Sect. 4.3.

## 5.2 Anomalous events

Another view of the Arctic amplification is the change in the date of early snow melt onset, as inferred from passive microwave satellite observations over sea ice (Markus et al., 2009). This date represents the first day under melting conditions and is plotted over both the climatological period and for 2017 relative to the climatological period in Fig. 12.

Arctic-wide, the climatology 1979–2016 shows a continuous increase in the date of melt onset with latitude from around ordinal date 100 (April 10) in the outer regions to later than 160 (June 9) in the central Arctic Ocean (Fig. 12a). In the Fram Strait, the climatological transition zone from early to late onset of melt is much more compressed, starting around ordinal date 140 (May 20), with a tongue of about 10 days later onset in the area of the West Spitsbergen Current west of the Yermak Plateau.

In 2017, the snow melt started 10–30 days earlier than normal in the eastern vicinity of the Northeast Water Polynya (cf. discussion in Sect. 4.3; Fig. 12b). This early onset is also found in other recent years (not shown). Conversely, the snow on sea ice both west and east of this area started melting 10–30 days later in 2017 relative to climatology.

Building on Fig. 5a (cf. discussion in Sect. 3.3), Fig. 13 shows the occurences, duration, and intensity of MCAOs and WAAs over the ACLOUD/PASCAL comparison period May 23 – June 26, 1998–2017. As for the AO and AD indices in Fig 11, we present the most recent and relevant 20 years in Fig. 13 even though calculations were made over the climatological period 1979–2017.

We identified six MCAO events over the 20 year ACLOUD/PASCAL comparison period (Fig. 13a). These lasted from two to eight days and had intensities of 2.5–5.6 K. In 2017, one MCAO event took place (cf. Fig. 5a), which was in the upper part of the climatological range, lasting 7 days with an intensity of 4.7 K. While remarkable for the season with convective rolls and cloud streets in satellite images, this MCAO is weak compared to cold season MCAOs, when indices reach more than 10 K (Fletcher et al., 2016; Chechin and Lüpkes, 2017).

Warm air advections are more common in late spring, with 21 events recognized over the ACLOUD/PASCAL comparison period (Fig. 13b). Duration and strengths of these reached up to 12 days and 14 K, respectively, although the majority lasted less than 8 days and were weaker than 9 K. In 2017, two moderate WAAs took place (cf. Fig. 5a). These lasted 6 and 7 days and had intensities of 9.1 K to 10.3 K, respectively.





### 5.3 Other campaigns

The few observations in the ACLOUD/PASCAL region partly explains the motivation for the field campaigns. Parodoxically, this also makes it hard to compare the data shown in this manuscript to other studies. Nevertheless, with differences in years, seasons, locations, and set-ups taken into account, such a comparison is still relevant for understanding the rapidly changing Arctic climate system.

SHEBA (cf. Fig. 1a) was the first field campaign to include a full year of Arctic measurements (Uttal et al., 2002). Taking place from October 1997 to October 1998, its main objective was to advance the understanding of the coupled ocean-ice-atmosphere processes in models. While taking place in the ice pack of the Beaufort Sea, some comparisons to ACLOUD/PASCAL can still be made.

During May and June 1998, temperature inversion heights of about 200–700 m and persistent cloudiness (80–100 %) characterized the SHEBA ice camp (Uttal et al., 2002). Over the same months in 2017, we observed inversion heights both shallower (about 100 m) and deeper (about 1400 m) north of Svalbard (Fig. 4a), along with cloudy conditions in the whole region (Fig. 10). Like in Ny-Ålesund (Fig. 2b), the snow melt season began May 29 and ended during the first half of June during SHEBA. Overall, the spring at the SHEBA station was slightly warmer than average (Uttal et al., 2002), while being close-to-average in Ny-Ålesund in 2017 (Fig. 2b). Taking place 19 years later, the relatively colder temperatures observed during ACLOUD/PASCAL could seem counterintuitive. However, while SHEBA temperatures were compared to the 1979–1993 time period (Martin and Munoz, 1997), the climatology for Ny-Ålesund temperatures in Fig. 2b is 1993–2016. Hence, the latter period contains the Arctic amplification period (Serreze and Barry, 2011).

The drifting ice station TARA took place in the central Arctic Ocean during the International Polar Year September 2006 to September 2007 (cf. Fig. 1a) and thus within the trend of rapidly rising Arctic temperatures (Vihma et al., 2008). Nevertheless, the summer (as defined by snow/sea ice temperature) started later than at SHEBA: on June 9 compared to May 30. Along with warmer and moister mean profiles April–August during SHEBA, this might be a result of the more northerly location of Tara (Vihma et al., 2008). While mostly north of SHEBA, mean profiles during ACLOUD/PASCAL were typically warmer and moister (Fig. 7), plausibly due to the more synoptic active region of the latter campaign (Serreze et al., 1997).

The Swedish icebreaker *Oden* has been regularly deployed in the Atlantic sector of the Arctic Ocean over the last decades. It was used for the two expeditions AOE-96 in July–September 1996 and AOE-2001 in June–August 2001, as well as the more recent ASCOS expedition in August–September 2008 (cf. Fig. 1a; Tjernström et al., 2012). While their main focus was on the late summer season, comparisons to the more recent ACLOUD/PASCAL campaign are still relevant due to the more southerly location and stronger influence of the Arctic amplification of the latter.

ASCOS was dominated by anticyclonic atmospheric circulation, while cyclonic circulation prevailed during AOE-96 and AOE-2001 (Tjernström et al., 2012). During the ACLOUD/PASCAL measurement period, we found strong daily variability (Fig. 5b), with cyclonic and anticyclonic circulation governing CP and WP, respectively (Figs. 8a and 8c). Nevertheless, as was the case here (Fig. 6), marked differences in air flow regimes were also observed during ASCOS (Fig. 9 in Tjernström et al., 2012).





Relative to SHEBA, AOE-2001 and ASCOS (Tjernström et al., 2012), near-surface air temperatures while in the ice were primarily $2°$ C lower at Polarstern during ACLOUD/PASCAL (Fig. 2b). However, care should be taken in the comparison due to the different measurement heights (5–20 m compared to 29 m) and days (21–64 days compared to 11 days), in addition to season (late compared to early summer). Similar to the AOE-96, SHEBA, AOE-2001, and ASCOS campaigns (Tjernström

et al., 2012), we observed inversions and these mostly in the lowest kilometer in almost all profiles above Polarstern (Fig. 4). Of the three mean profiles in Fig. 7, NP (i.e., the last and most representative key period) corresponds best to the profiles from AOE-96, SHEBA, AOE-2001, and ASCOS (Figs. 15a and 15b in Tjernström et al., 2012).

Most comparable to ACLOUD/PASCAL would be the N-ICE2015 expedition (cf. Fig. 1a). This took place in the sea ice north of Svalbard and included sea ice drift measurements in winter and spring 2015 (Granskog et al., 2016). May and June

temperature values and variability were similar in 2015 (Fig. 3b in Cohen et al., 2017) and 2017 (Fig. 2b here), with mostly lifted temperature inversions and surface-based humidity inversions (Fig. 3 in Kayser et al., 2017 compared to Fig. 4 here). As observed during SHEBA, Tara, and N-ICE2015 (Cohen et al., 2017), the summer appeared to begin around the first week of June during ACLOUD/PASCAL (Fig. 2b here).

In general, ACLOUD/PASCAL was to a low degree influenced by synoptic cyclones (as indicated by the few marked changes

in Figs. 2b and 2c in association with the changes in Fig. 2a). This is different to the situation during N-ICE2015, in which a persistent and anomalous low pressure centered over the Barents Sea dominated the according season (Cohen et al., 2017). In 2015, this meant more abrupt shifts in cloud cover due to the associated cyclonic circulation (Cohen et al., 2017; Graham et al., 2017; Kayser et al., 2017); in 2017, we observed mostly cloudy conditions in the dominating anticyclonic circulation (Figs. 5b and 10). We also found no marked precipitation events to follow from pressure drops (Figs. 2a and 2c here compared to Fig.

3a in Cohen et al., 2017).

## 6   Summary and concluding remarks

This manuscript provides an overview of the synoptic development during the ACLOUD airborne and PASCAL ship-based field campaigns, which took place near Svalbard from May 23 to June 26, 2017. This development is presented using near-surface and upper-air meteorological observations, satellite, and model data.

Time series of the data from Ny-Ålesund (at $79°$ N, $12°$ E) and Polarstern ocean-crossing (in the Nordic Seas north of the Arctic Circle) and ice-attached locations (at about $82°$ N, $10°$ E) during the 35 day measurement period are presented and compared to the long-term near-surface and radiosonde measurements conducted in Ny-Ålesund. Additionally, we computed the marine cold air oubreak (MCAO) index and the circulation weather type (CWT) classification, and compared the former to its climatology of the region.

Relative to the long-term averages, we identified three key periods: (1) a cold period (CP; May 23–29; 7 days), (2) a warm period (WP; May 30 – June 12; 14 days), and (3) a normal period (NP; June 13–26; 14 days). These were characterized by (1) cold and dry Arctic air advected from the north, (2) warm and moist maritime air transported from the south and east, and (3) close-to-average temperate and moist air from a mixture of regions (but dominanted by adiabatically warmed air from





the west). The resulting sea ice drift caused an anomalous southerly sea ice edge in the Fram Strait, packing of the ice edge and opening of the Northeast Water Polynya, respectively. Associated with the cold and dry Arctic air flow, low-level stratus clouds prevailed over the open ocean in CP, while the warm air advections were connected with complex, partly high-reaching cloud systems in WP. NP showed a mix of both conditions. Thus, relative to the long-term observations, we found short-term
variability in atmospheric circulation more visible than the long-term background forcing of the Arctic amplification.

Our focus was limited to the North Atlantic sector of the Arctic. Hence, the results presented here do not necessarily translate to the whole Arctic climate system. For this, the regional differences are too large (e.g., Serreze et al., 2011; Cavalieri and Parkinson, 2012; Koyama et al., 2017). For example, sea ice coverage in the region was anomalously large and far south as a result of the strong southward drift during CP and, albeit weaker, still southward drifts during WP and NP.
Nevertheless, in the background of the sparsely observed Arctic region, the extensive ACLOUD/PASCAL campaign offers unique measurements, with observations over the ocean, sea ice, and snow, and of the whole tropospheric column. Most measurements performed during ACLOUD/PASCAL will be continued in the framework of the Multidisciplinary drifting Observatory for the Study of Arctic Climate (MOSAiC; IASC, 2016), including a one-year ice drift of Polarstern and several aircraft- and ground-based campaigns. Thus, while MOSAiC will strongly benefit from the results and experiences we have
gained from ACLOUD/PASCAL, the campaigns together are anticipated to considerably improve the understanding of the cloud-related processes in the Arctic atmosphere, as well as the ocean-ice-atmosphere interaction from turbulent and radiative energy fluxes. Ultimately, this will strengthen synoptic forecasting in weather and climate models, beneting actors beyond the scientific community.

*Code availability.* TEXT

*Data availability.* The surface-based measurement data used in this manuscript are available through the information system PANGAEA (Maturilli, 2017b, a; Schmithüsen, 2017, 2018), hosted by AWI, Helmholtz Center for Polar and Marine Research and the Center for Marine Environmental Sciences (MARUM), UB, and the MET Norway web portal eKlima. Satellite data are accessible through UB (Spreen et al., 2017), NSIDC (Fetterer et al., 2018), OSI SAF (Lavergne et al., 2010), and the European Organisation for the Exploitation of Meteorological Satellites (EUMETSAT; EUMETSAT, 2017). Finally, reanalysis and analysis data used in this study can be obtained from ECMWF (Dee
et al., 2011)

*Code and data availability.* TEXT

*Sample availability.* TEXT





**Appendix A**

*Author contributions.* E. M. Knudsen led the coordination and design of the study, analyzed data for and plotted Figs. 2, 8, A3, and A4, and wrote all sections except for Sect. 1 based on the input from the co-authors. B. Heinold helped in the coordination and the design of the study, and also analyzed data, plotted, and provided descriptive text for Figs. 1, 5, 11, 13, and A2. S. Dahlke helped in the design of the study, and

also analyzed data, plotted, and provided descriptive text for Figs. 3, 4, and 7. H. Bozem and D. Kunkel analyzed data, plotted, and provided descriptive text for Fig. 6. S. Crewell and M. Mech analyzed data, plotted and provided descriptive text for Figs. 10 and A5. G. Heygster analyzed data, plotted, and provided descriptive text for Figs. 9 and 12. M. Maturilli and H. Schmithüsen provided data from Ny-Ålesund and Polarstern, respectively. A. Rinke wrote Sect. 1. C. Lüpkes provided Fig. A1. In addition to the other authors, A. Ehrlich, A. Macke, and M. Wendisch evaluated the study and manuscript.

*Competing interests.* The authors declare that they have no conflict of interest.

*Disclaimer.* TEXT

*Acknowledgements.* We gratefully acknowledge the support from the Transregional Collaborative Research Center (TR 172) "ArctiC Ampli-fication: Climate Relevant Atmospheric and SurfaCe Processes, and Feedback Mechanisms (AC)[3]", which is funded by the German Research Foundation (Deutsche Forschungsgemeinschaft; DFG). We thank T. Marke for providing data used in Fig. 5b and D.C. Strack for data used

in Fig. 10, as well as M. Gerken for preparing plots in Fig. 9 and C. Patiliea for plots in Fig. 12. M. Kayser should also be mentioned for valuable comments on an earlier version of this manuscript. M. Rautenhaus is acknowledged for providing the Mission Support System (MSS; Rautenhaus et al., 2012) for flight planing during ACLOUD, as well as J. Ungermann and R. Bauer for technical support.





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





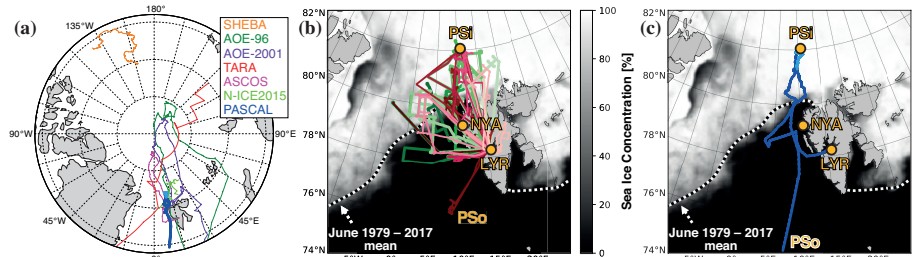

**Figure 1.** Overview of the (a) Arctic and (b,c) the ACLOUD/PASCAL region. (a) Tracks of the icebreaker *Polarstern* during PASCAL (blue) and previous Arctic ship-based campaigns (orange to light green; see Sect. 5.3 for description). For the former, dark and bright colors indicate ocean-cruising (PSo; May 30 – June 5 and June 17–18, 2017) and ice-attached (PSi; June 6–16, 2017) positions, respectively. (b) Tracks of the aircraft Polar 5 (green) and Polar 6 (red) flights during ACLOUD May 23 – June 26, 2017, with later dates in brighter colors. (c) Track of PSo cruise and PSi position (blue). In (b) and (c), codes represent Longyearbyen (LYR), Ny-Ålesund (NYA), PSo entering the ACLOUD/PASCAL region, and PSi, while the shading and the dashed line represent the average sea ice concentration over the ACLOUD/PASCAL measurement period May 23 – June 26, 2017, and edge (defined by 15 % concentration) June 1979–2017, respectively (see Sect. 2.3 for data explanation).




**(a)** *Near-surface pressure and 850-hPa wind*

**(b)** *Near-surface temperature and snow melt season*

**(c)** *Integrated water vapor and precipitation*

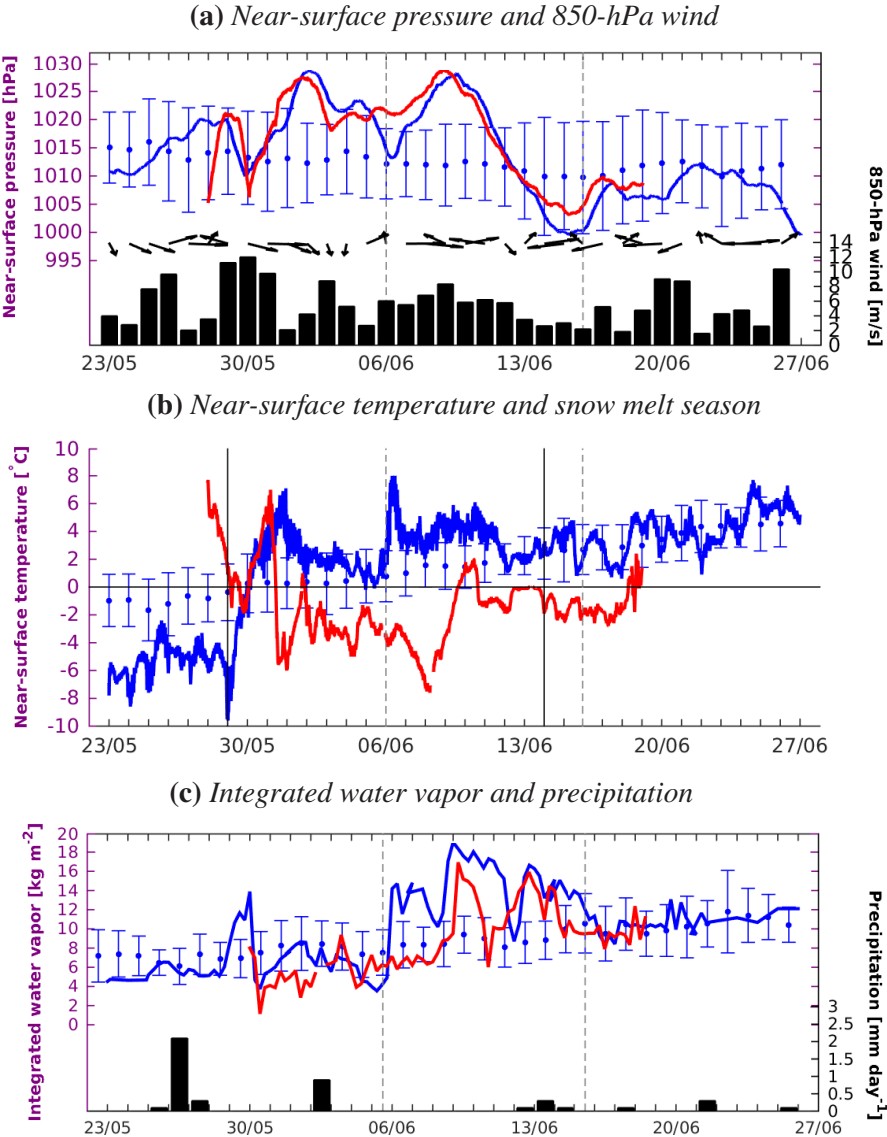

**Figure 2.** (a) Near-surface pressure (graphs) and 850 hPa horizontal wind (bars for speed, vectors for direction), (b) near-surface air temperature (graphs) and snow melt season (solid vertical lines), and (c) vertically integrated water vapor (graphs) and precipitation (bars) measured at Ny-Ålesund (blue and black) and Polarstern (red) over the ACLOUD/PASCAL measurement period May 23 – June 26, 2017. Dots and intervals indicate daily average and standard deviation, respectively, over the Ny-Ålesund long-term period 1993–2016. Dashed vertical lines distinguish the Polarstern ocean-crossing periods from the ice-attached period (June 6–16).




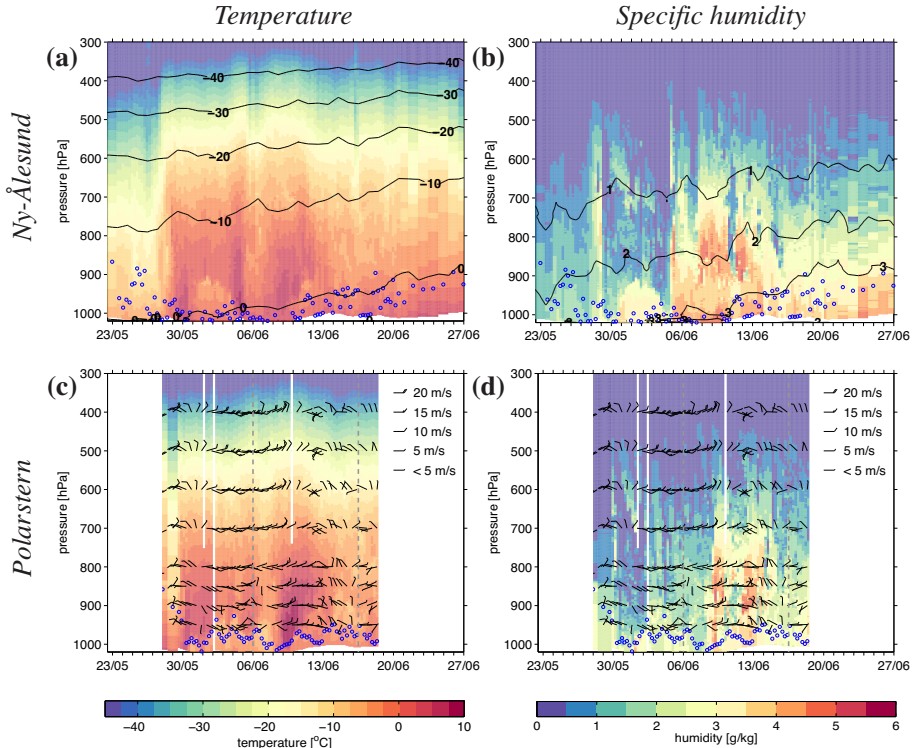

**Figure 3.** Vertical profiles of [left column] temperature and [right column] specific humidity measured at [upper row] Ny-Ålesund and [lower row] Polarstern over the ACLOUD/PASCAL measurement period May 23 – June 26, 2017. Blue circles indicate the height of the atmospheric boundary layer. Black contour lines in upper row panels represent the respective 1993–2016 average, while black arrows in lower row panels represent 2017 values of wind speed and direction. Dashed vertical lines distinguish the Polarstern ocean-crossing periods from the ice-attached period (June 6–16).



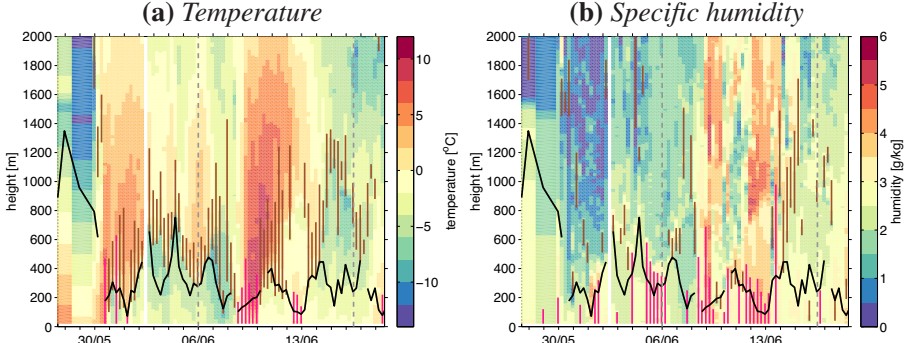

**Figure 4.** Vertical profiles of (a) temperature and (b) specific humidity measured at Polarstern over the PASCAL measurement period May 28 – June 18, 2017. Pink and brown vertical lines indicate the vertical extent of the lowermost surface-based and lifted inversion, respectively, while black contour lines indicate the atmospheric boundary layer height corresponding to the blue circles in Fig. 3. Dashed vertical lines distinguish the Polarstern ocean-crossing periods from the ice-attached period (June 6–16).





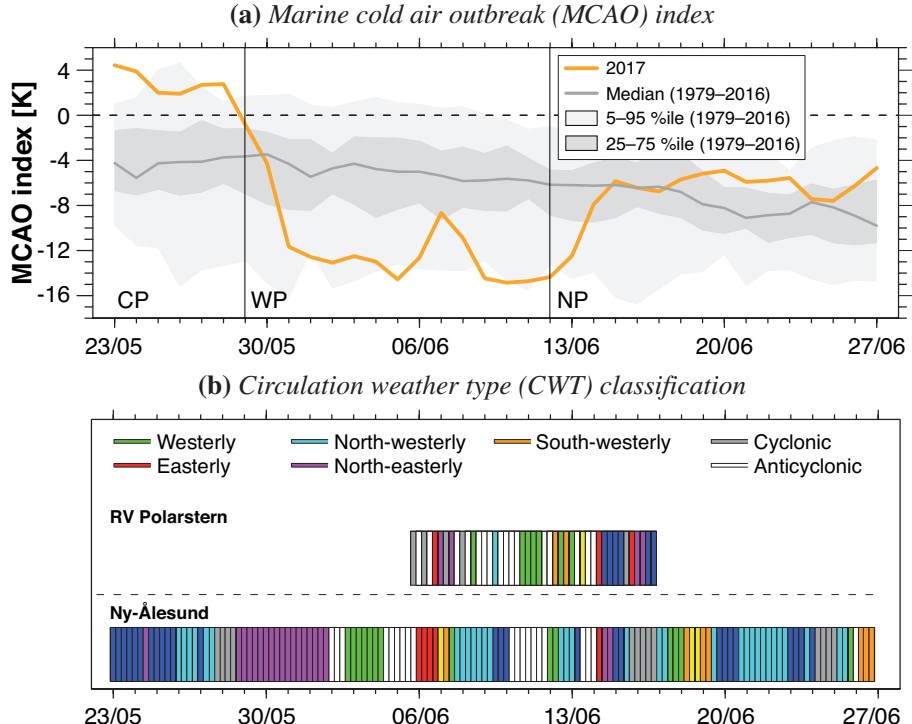

**Figure 5.** (a) Marine cold air outbreak (MCAO) index for the eastern Greenland Sea (75.00–80.25° N, 4.50–10.50° E) and (b) circulation weather type classification for Ny-Ålesund (77.25–81.00° N, 9.75–14.25° E) and Polarstern (80.25–84.00° N, 7.50–12.75° E) over the ACLOUD/PASCAL measurement period May 23 – June 26, 2017, based on ERA-I data. In (a), the gray median line and percentile shading refer to the climatology over 1979–2016, while the black vertical lines separate the three key periods CP, WP, and NP in 2017 defined in Sect. 4.

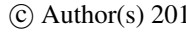



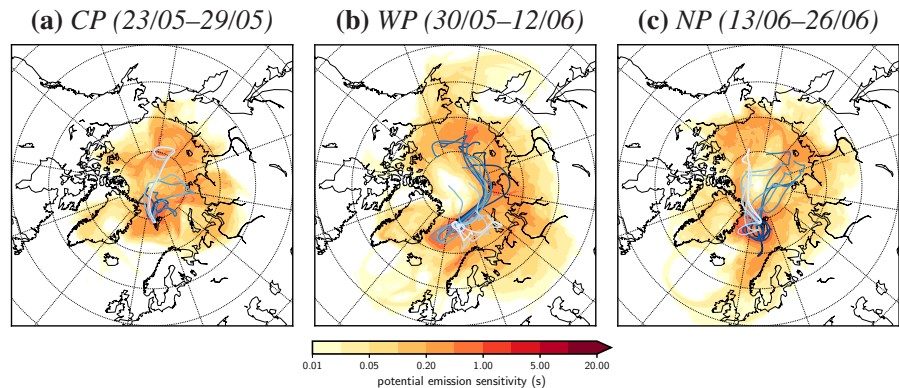

**Figure 6.** Ranges of continuously daily potential emission sensitivities (shading) and daily center of mass backward trajectories to Ny-Ålesund, with later masses in brighter colors (trajectories) for each ACLOUD/PASCAL key period from FLEXPART. The key periods are defined as (a) the cold period (CP), (b) the warm period (WP), and (c) the normal period (NP).





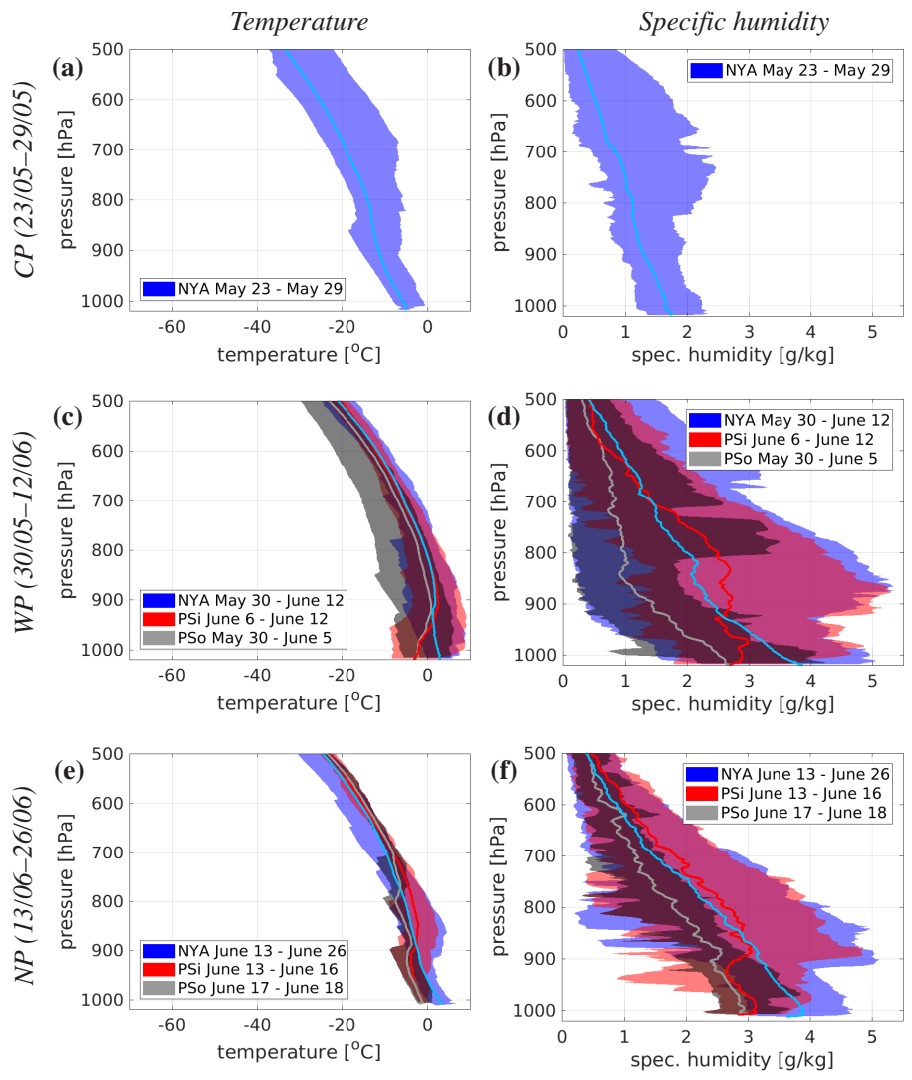

**Figure 7.** Average (graphs) and ranging (shading) vertical profiles of [left panels] temperature and [right panels] specific humidity for each ACLOUD/PASCAL key period from Ny-Ålesund (blue) and Polarstern (red for ice-attached dates, gray for cruising dates). Key periods are defined as [first row] the cold period (CP), [second row] the warm period (WP), and [third row] the normal period (NP).

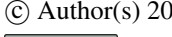



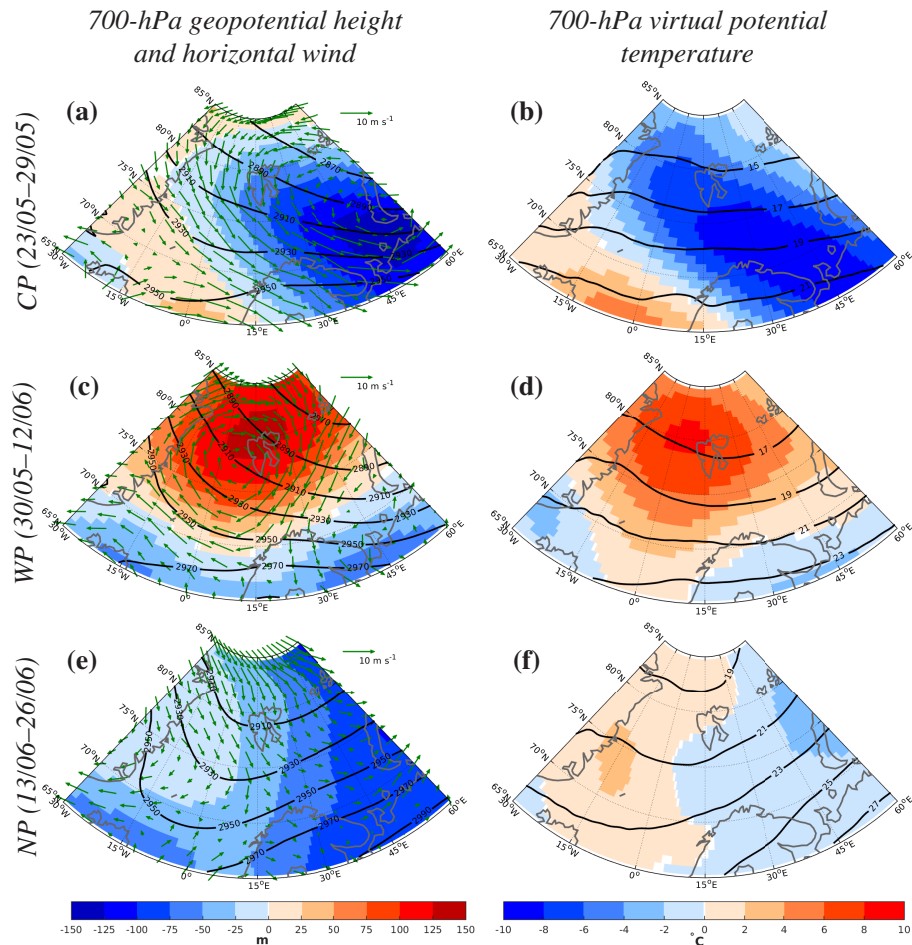

**Figure 8.** Climatologies (1979–2016; contours) and anomalies relative to climatologies (2017 minus 1979–2016; shading) of 700 hPa [left panels] geopotential height with key period median horizontal wind (vectors) and [right panels] virtual potential temperature for each ACLOUD/PASCAL key period based on ERA-I data. Key periods are defined as [first row] cold period (CP), [second row] warm period (WP), and [third row] normal period (NP).





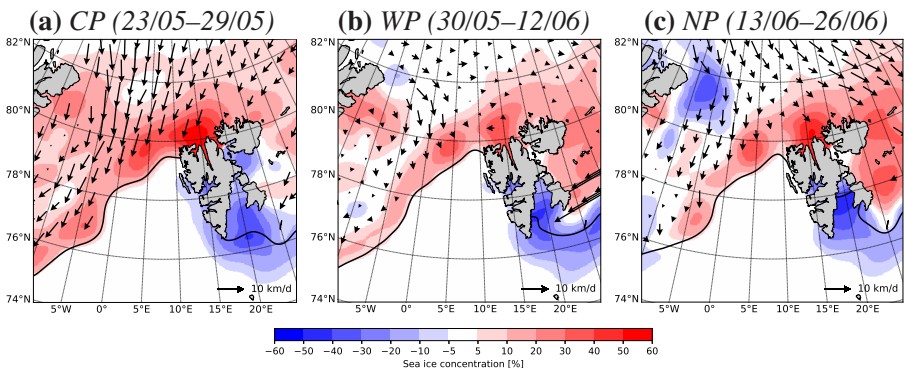

**Figure 9.** Anomalous sea ice concentration relative to climatologies (2017 minus 1979–2016; shading) and average sea ice drift (2017; vectors) for each ACLOUD/PASCAL key period from UB, NSIDC, and OSI SAF. The key periods are defined as (a) the cold period (CP), (b) the warm period (WP), and (c) the normal period (NP). White shading south of the 2017 sea ice edge (line) indicates open water.





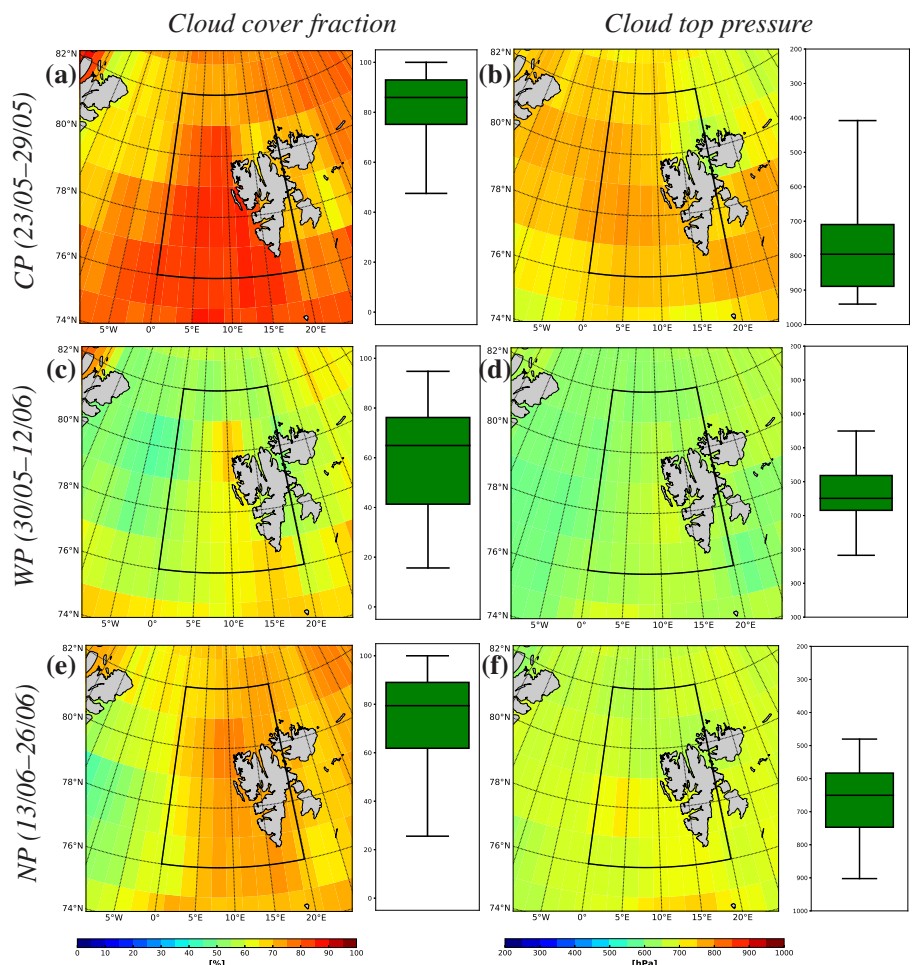

**Figure 10.** Average [left column] cloud cover fractions and [right column] cloud top pressures for each ACLOUD/PASCAL key period from IASI. Box plots represent averages over the central ACLOUD/PASCAL region (76–82° N, 0–20° E; black boxes in map panels), with ticks indicating the 5 and 95 percentiles. The key periods are defined as [first row] the cold period (CP), [second row] the warm period (WP), and [third row] the normal period (NP).



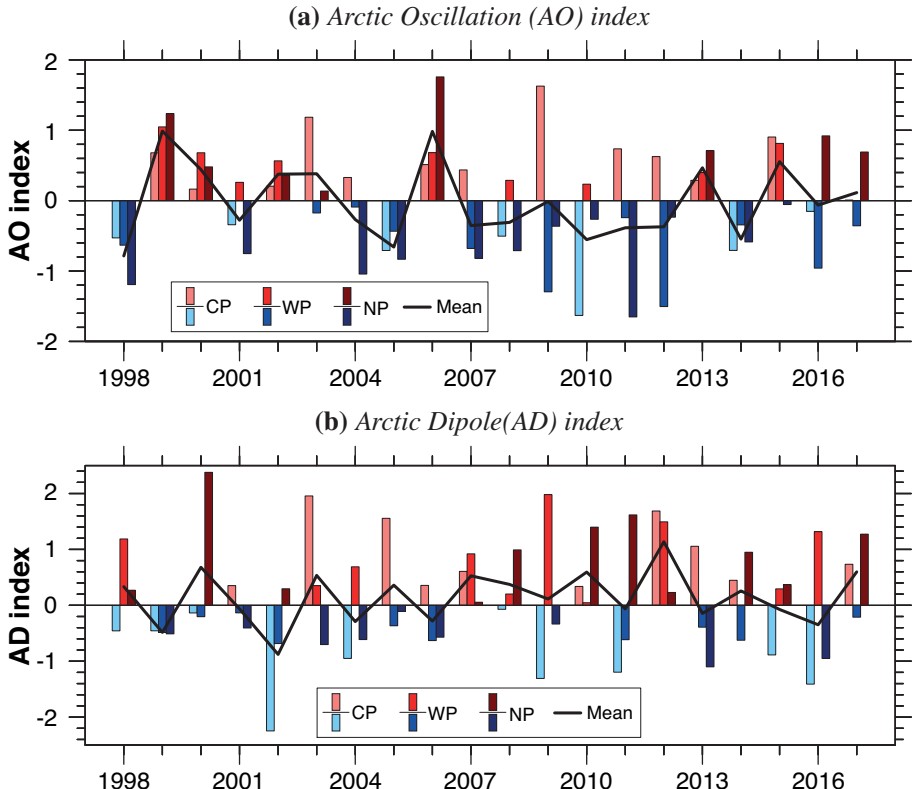

**Figure 11.** Time series of (a) Arctic Oscillation (AO) and (b) Arctic Dipole (AD) indices for the Northern Hemisphere over the ACLOUD/PASCAL comparison period 1998–2017 based on ERA-I data. Lines and bars indicate campaign and key period averages, respectively. These are defined as May 23 – June 26 (mean), May 23–29 (the cold period; CP), May 30 – June 12 (the warm period; WP), and June 13–26 (the normal period; NP), respectively.





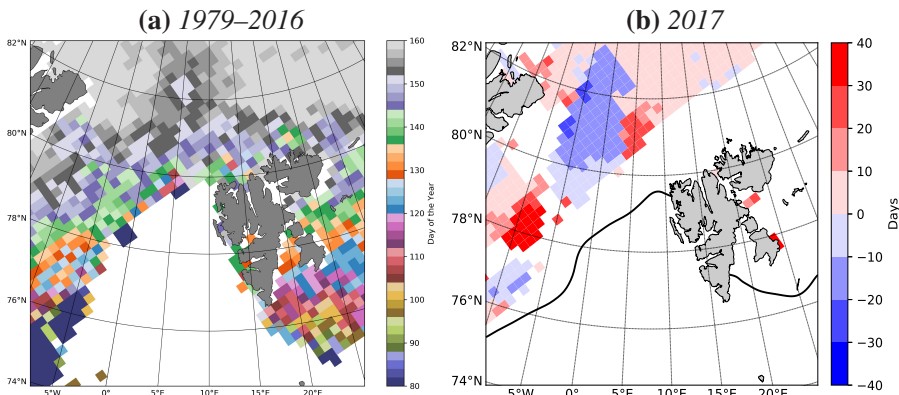

**Figure 12.** (a) Climatology (1979–2016) and (b) anomaly relative to the climatology (2017 minus 1979–2016) of snow melt onset date based on NASA GSFC data. In (b), white shading south of the 2017 sea ice edge (line) indicates open water.





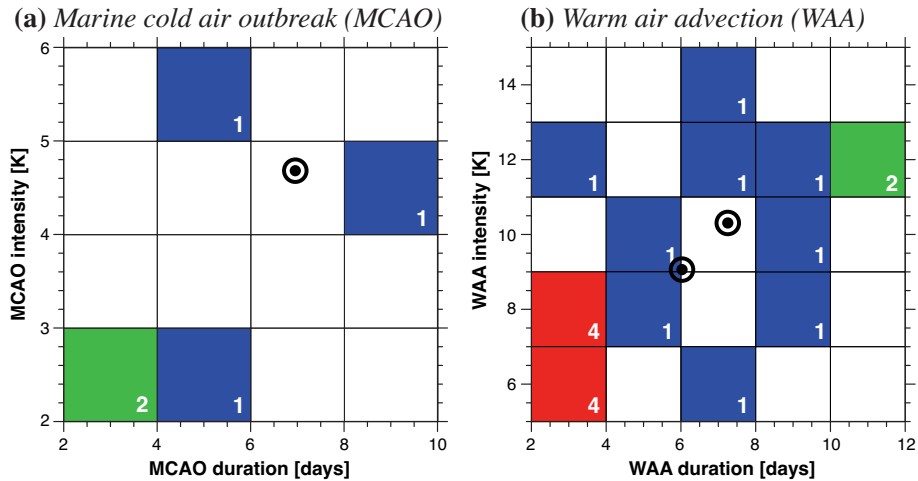

**Figure 13.** (a) Marine cold air outbreak (MCAO) and (b) warm air advection (WAA) durations and intensities for the eastern Greenland Sea (75.00–80.25° N, 4.50–10.50° E) over the ACLOUD/PASCAL comparison period May 23 – June 26, 1998–2017, based on ERA-I data. White numbers represent the number of MCAO and WAA events over 1998–2016 (also highlighted by different colors), while black dots represent 2017 events.





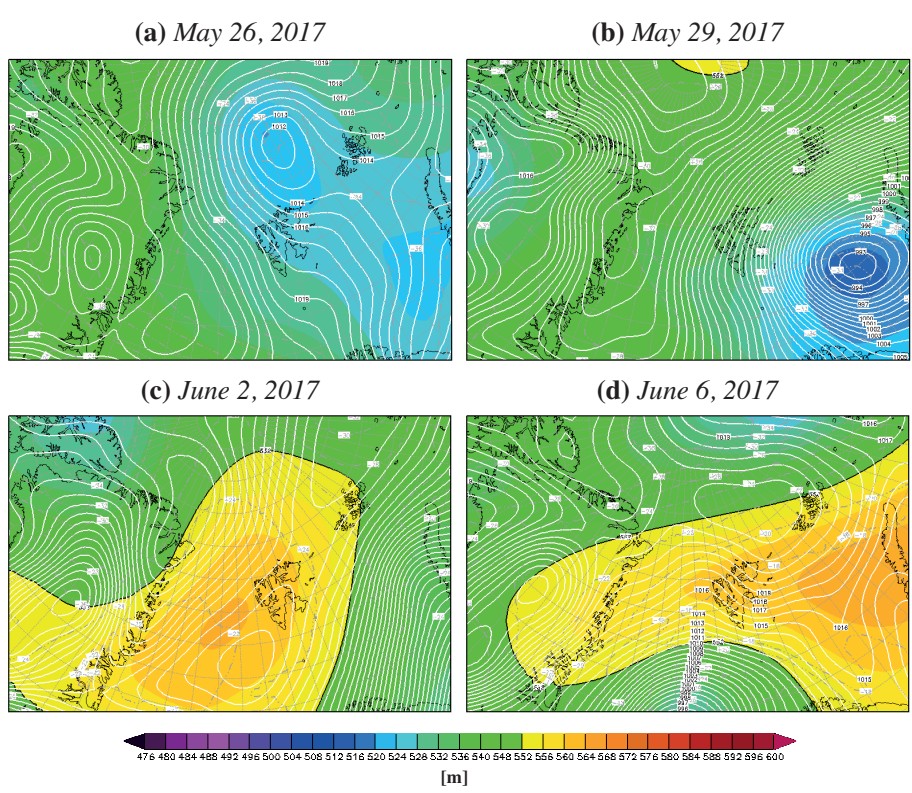

**Figure A1.** Sea level pressure (in hPa; white contours), 500 hPa temperature (in °C; grey contours), and 500 hPa geopotential height (in m; shading) for (a) May 26, (b) May 29, (c) June 2, and (d) June 6, 2017, from www.wetterzentrale.de.





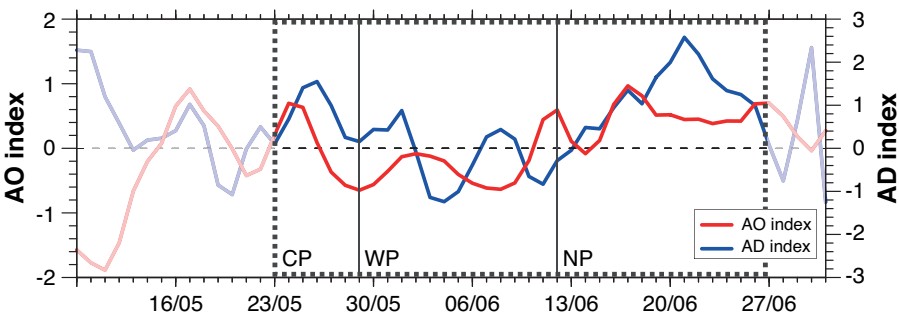

**Figure A2.** Time series of daily Arctic Oscillation (AO) and Arctic Dipole (AD) indices over the ACLOUD/PASCAL extended period May 9 – July 1, 2017, based on ERA-I data. Dotted boxes indicate the ACLOUD/PASCAL measurement period May 23 – June 26, while vertical lines separate the three key periods CP, WP, and NP defined in Sect. 4.





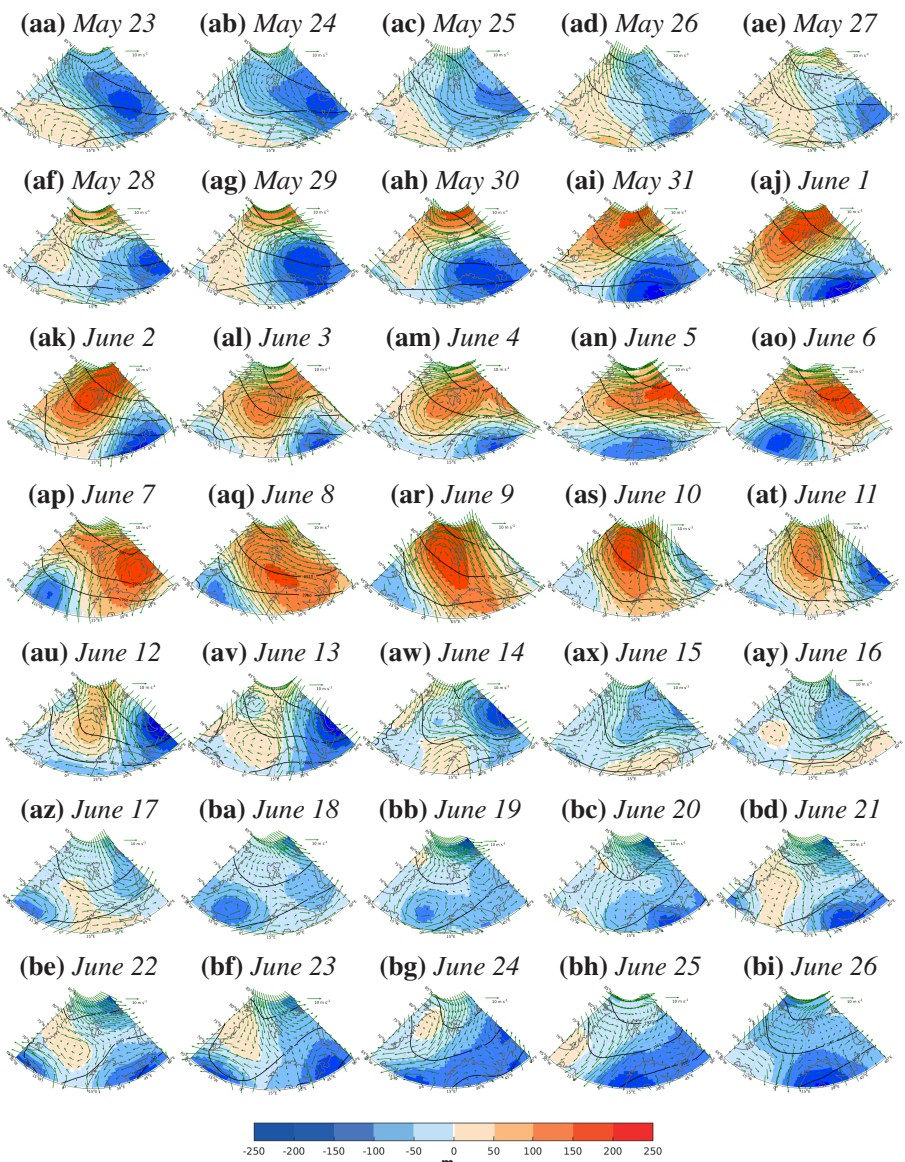

**Figure A3.** Daily climatologies (1979–2016; contours) and anomalies relative to climatologies (2017 minus 1979–2016; shading) of 700 hPa geopotential height with horizontal wind (vectors) over the ACLOUD/PASCAL measurement period May 23 – June 26, 2017, based on ERA-I data.





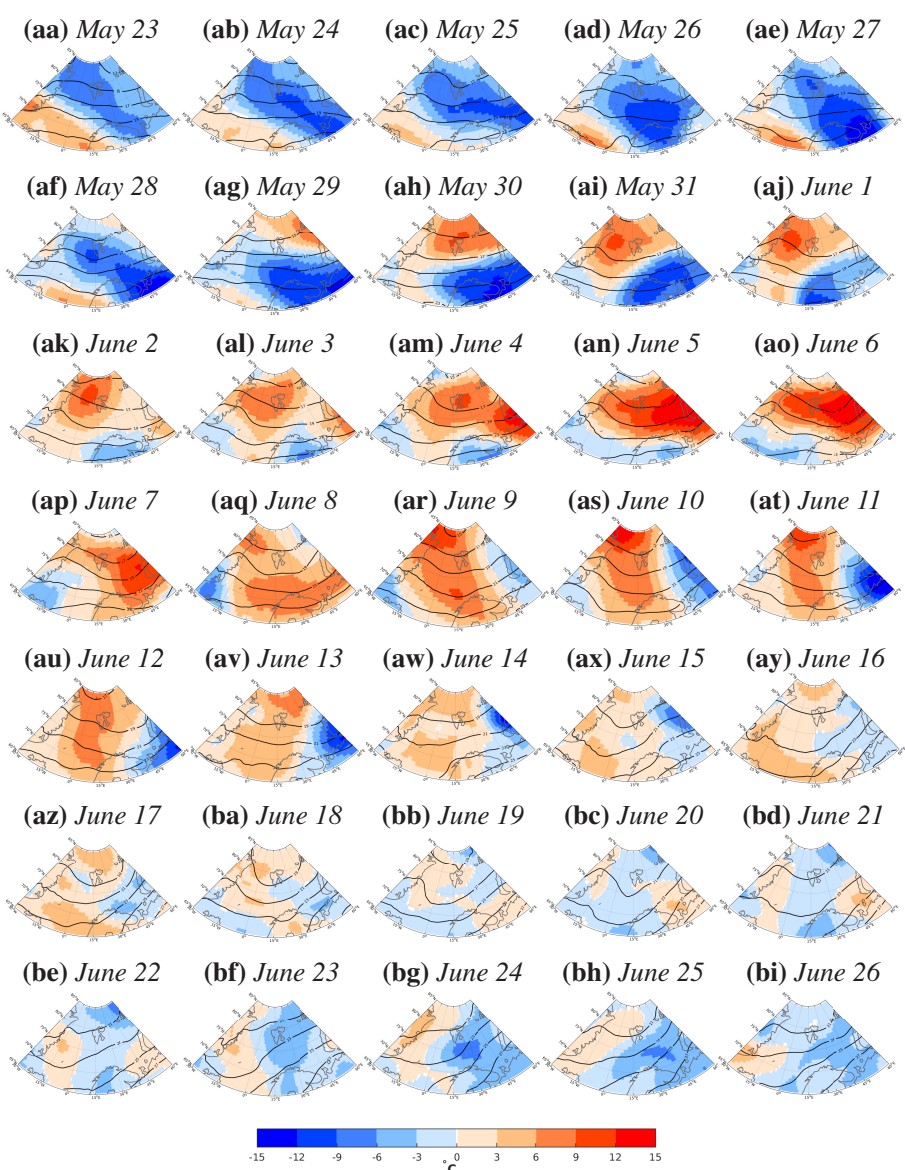

**Figure A4.** Daily climatologies (1979–2016; contours) and anomalies relative to climatologies (2017 minus 1979–2016; shading) of 700 hPa virtual potential temperature over the ACLOUD/PASCAL measurement period May 23 – June 26, 2017, based on ERA-I data.



**(a)** *Cloud cover fraction*

**(b)** *Cloud top pressure*

**Figure A5.** Cloud (a) cover fraction and (b) top pressure averaged over the central ACLOUD/PASCAL region (76–82° N, 0–20° E; black boxes in Fig. 10) over the ACLOUD/PASCAL measurement period May 23 – June 26, 2017, based on IASI data. Ticks indicate the 5 and 95 percentiles.