# Peer review of "Meteorological conditions during the ACLOUD/PASCAL field campaign near Svalbard in early summer 2017"

_Atmospheric Chemistry and Physics, 2018_

## Referee Comment (RC1) · L. Cohen (Referee) · 7 Jul 2018

**Review comments for: "Synoptic development during the ACLOUD/PASCAL field campaign near Svalbard in spring 2017"**

This manuscript characterizes the synoptic conditions during two concurrent campaigns, one ship-based and one airplane-based, conducted in May and June 2017 in the region north of Svalbard. The analysis aims to provide climatological and synoptic context for the interpretation of subsequent analysis of cloud and aerosol data. During the campaign, three distinct periods are classified and described using various data sources. This manuscript shows how synoptic variability is related to the variability in surface observations, atmospheric profiles, and clouds using the campaign-based data as well as Ny Ålesund observations, reanalysis and satellite data. The analysis also shows that sea ice conditions are largely driven by large-scale atmospheric circulation during this period. The synoptic variability is also put into a climatological context using Ny Ålesund records and large-scale atmospheric indices.

This manuscript provides relevant and significant contribution to the scientific community. I recommend the manuscript for publication with some relatively minor modifications. These are described in the general and specific comments below.

**General comments:**

In general, throughout the manuscript, the text could be more concise by focusing on the most relevant points from each figure rather than simply describing everything about it. This would help the reader understand which features or points are most important. There are more specific notes below in the Specific comments.

Several figures could be eliminated or combined for clarity. These are pointed out in the specific comments below.

The discussion of synoptic settings (Sec 3.1) needs some clarity. It may help to highlight specific periods on Figure 2, but in general, the narrative style is a bit difficult to read, so perhaps also shortening it a bit would help. Some suggestions are made in the Specific comments below.

It seems the conclusions could be stated more explicitly in order to express clearly what the conclusions from this analysis are and why they are relevant. For example, this work shows that 1) the synoptic variability in this region drives much of the surface meteorology, atmospheric profiles, and cloud distribution; 2) the large-scale atmospheric circulation drives sea ice distribution during this period; 3)  Ny Ålesund is fairly representative of the mid-to upper troposphere in this region; 4) understanding the boundary layer variability is important for understanding surface observations; 5) synoptic variability is linked to the large-scale atmospheric variability. These are just a few key results that are not included in the conclusions.

Finally, the text could use another round of editing to improve the language and readability.

**Specific comments:**

P1 L5: Not sure what is meant by "classical" near-surface… observations.

P4 L23 & P5 L1: Include "Vaisala" before RS41 and RS92

P5 L1-3: These first two sentences are poorly worded. Please clarify.

P5 L34: Explain what "level 2 cloud cover" is.

P6 L2-5: Sentence is poorly worded.

P6 L14: It should be noted that the Ny Ålesund radiosondes are assimilated into ERA-Interim. Were the extra sondes launched during this campaign also assimilated?

P7 L2: Describe the synoptic situation that is driving this cold air outbreak. This could be shown using the synoptic maps that are included in Fig A1.

P7 L3-7: This paragraph is not at all clear. Please rework to clarify.

P7 L8: "The variable wind direction over Svalbard finally caused the highest precipitation…" It cannot be said that wind direction caused precipitation, please explain this more clearly.

P7 L10: In this discussion, it would be helpful to highlight the periods in the timeseries (in Fig 2) that you are describing.

P7 L13: "A few days later…" Its not clear which days are being referred to here.

P7 L18: How do you know there was an atmospheric river? It is not apparent in Fig A1c that it would it be coming from the east. This should be clarified.

P7 L26: This paragraph is not clear. This first mentions beginning of snow melt season, but then switches to first snow-free day and albedo. Please make this more clear.

P9 L14-31: The discussion of lifted inversions does not add much to the overall analysis. Perhaps it could just be mentioned briefly or merged into the discussion of Figure 3?

P10 L27-30: It doesn't seem that the CWTs add much to the discussion or understanding of the synoptics or large scale circulations. In fact it only raises questions (eg. CWTs for Ny Ålesund and Polarstern are more different that one might expect!). I think this discussion (and Fig 5b) could be left out altogether.

P11-12: It is not clear that the back trajectories are adding much to the analysis, and in this section there are some issues with these trajectories that are not addressed. The main issues are 1) the very large uncertainties that exist for going back 10 days (only 5 days is often used for this reason) should be addressed; 2) spatially averaging trajectories with large variability can be very misleading and this should be acknowledged; and 3) it seems overly complicated to use PES instead of a simple residence time (perhaps you could use HYSPLIT instead of FLEXPART)?

P12-13: This section discussing Fig 7 could be much more concisely written. What are the key features of Fig 7? Rather than describing every feature of the plots, just focus on 2 or 3 important points.

P13 L26: Figures A3 & A4 are not mentioned further in the text, and can therefore be eliminated.

P16 L22: Very minor point, but perhaps Section 5.1 should be titled "Large-scale indices" since it does not actually deal with atmospheric dynamics in any way.

P18 L20: Not sure you need Figure 13; I think the text description here is enough.

P19 L8: Make more explicit that SHEBA was in a completely different part of the Arctic and that there are considerable regional differences.

P19 L14-18: Not sure that any of this comparison to SHEBA is relevant due to different region and different climatology. I think it just confuses things, I would leave this out.

P19 L19-24: This paragraph is very confusing, since both Tara and SHEBA are mentioned in comparison to ACLOUD/PASCAL. Please make this more clear.

P20 L1-4: Again, the comparison here is irrelevant because of the different measurement heights and season.

P21 L1: Change to "The sea ice drift resulting from the large-scale atmospheric circulation caused…"

P21 L2: Delete "respectively".

**FIGURES:**

Fig 2a: The 850hPa windspeed presented as bar graph seems a bit odd, perhaps would be better as just points or line?

Figs 3 & 4: Perhaps these figures could be merged?

Fig 12a: The color scheme should have a smooth transition from day 80 to 160.

Fig 13: could be eliminated.

Fig A1: The grey contours cannot be seen, even when zoomed in a lot. Perhaps just choose the 2 parameters to show here, instead of 3.

Figs A3 & A4: could be eliminated.

---

## Referee Comment (RC2) · Anonymous Referee #2 · 21 Jul 2018

The manuscript presents a thorough description of synoptic-scale weather and sea ice conditions during a major field campaign in the Fram Strait region. Several complementary points of view are taken to characterize the synoptic conditions, and plenty of relevant analyses are carried out. The manuscript has a high potential to become an excellent paper, but substantial revisions are needed first.

Major comments

1. Introduction is good from the points of view of the Arctic climate system, climate modelling, small-scale physical processes, and field campaigns. However, much more attention should be put on synoptic-scale meteorology in the study region. What is

known and what are scientific challenges in the field?

2. The manuscript is entitled "Synoptic development during ACLOUD/PASCAL field campaign near Svalbard in spring 2017", but there is, in fact, very little attention to synoptic-scale dynamics. With the present title, a reader expects to learn, among others, about the mechanisms (such as baroclinic instability) affecting cyclogenesis and cyclolysis during the study period, and what was the role of the jet stream in steering the cyclone tracks. For example, on page 7, a careful description of the evolution of the synoptic situation is given, nicely linking synoptic conditions and point measurements during the campaign, but there is no deeper analysis on why the synoptic conditions developed as they did.

3. The results of the different analysis methods applied are not well linked to each other. I appreciate the analyses on in-situ and satellite observations, weather classification, air-mass distribution, atmospheric circulation and thermodynamics, clouds, and sea ice dynamics. However, the results should be put better together to summarize the synoptic development during the study period.

4. The manuscript includes a lot of information that will certainly be useful for those working with ACLOUD/PASCAL data, but its usefulness for a broader scientific community is not equally clear. To deserve publication in a high-level peer-reviewed journal, such as ACP, the manuscript should be useful for a broader community. This challenge can probably be met via a careful concern on topics 1 to 3 above and a proper discussion on the purpose of the manuscript.

Minor comments

Title: Consider replacing spring by early summer.

Page 1, line 10: . . . two cases of warm-air advection. Page 1, line 11: What is "westerly air"?

Page 2, line 6: Consider referring to Pithan and Mauritsen (2014)

Page 2, line 20: are the key

Page 3, line 21-22: Briefly describe the the "particularly marked climate changes".

Section 2.3: were any data collected on sea ice thickness?

Page 6, line 7: . . . reanalyses and operational analysis data . . .

Page 6, lines 9-10: Unclear sentence

Page 6, line 19: ECMWF operational analysis

Page 8, line 3: Clarify the sentence with ". . . surface cooling during warm-air advection . . .".

Page 9, line 19: objectively or subjectively chosen?

Page 11, line 26: Briefly describe what is the potential emission sensitivity.

Page 12: If airmass flows over Spitsbergen or Greenland, it first experiences adiabatic coolinf during the ascent, and then adiabatic warming during the descent. Your attention seem to be resctricted to the latter. Why? Further, do you catch the true adiabatic warming/cooling effects, if you integrate PES in the vertical?

Page 12, lines 26-28: Explain better. All radiosondes enter the free troposphere whether there are mountains or not.

Page 13, lines 8-12: The paragraph is very unclear. Clarify what is the effect of sea ice on the specific humidity in the lowermost 300 hPa.

Page 15, line 15: over the open ocean?

Section 4.4: Add brief characterization on fog conditions.

Line 17: Better explain the association with colder and wetter conditions in mid-latitudes. Which season you refer to? The all paragraph on AO makes more sense for winter than summer. In winter, Arctic cold-air outbreaks are typically associated

with cold and dry conditions in mid-latitudes.

Section 5.1. I wonder if AD, as defined by Wu et al. (2006) and Wang et al. (2009), is the best metrics to characterize meridional circulation patterns and atmospheric forcing on sea ice drift in the Arctic. See Vihma et al. (2012, GRL) for various weaknesses of AD. The Meridional Circulation Index (Francis and Vavrus, 2015, Env. Res. Lett.) may be a much more relevant metrics. Note that it can also be calculated on the basis of mean-sea-level pressure.

Page 21, lines 4-5: The sentence could be clarified. How would you characterize the long-term background forcing of the Arctic amplification on atmospheric circulation? It is fairly trivial that short-term variability in circulation is stronger than this forcing, which is not yet well known.

Page 21, lines 17-18: What do you meant by synoptic forecasting in climate models?

---

## Author Comment (AC1) · 2 Oct 2018

**Synoptic development during the ACLOUD/PASCAL field campaign near Svalbard in spring 2017**

Erlend M. Knudsen, Bernd Heinold, Sandro Dahlke, Heiko Bozem, Susanne Crewell, Irina V-Gorodetskaya, Georg Heygster, Daniel Kunkel, Marion Maturilli, Mario Mech, Carolina Viceto, Annette Rinke, Holger Schmithüsen, André Ehrlich, Andreas Macke, Christof Lüpkes, and Manfred Wendisch

**Set-up of response**

We thank the reviewers for their suggestions on the manuscript. With the changes explained below, we feel that the paper is strengthened compared to its first submission.

Following some of the comments in the review, we would like to point out that while not all aspects of the manuscript are key components for the story, they are nevertheless included as they will be important for upcoming papers from ACLOUD/PASCAL. To serve as references in these papers, we have included these aspects upon the request by the research community.

In the following, we go through each comment by the reviewers (reproduced here in gray text for your reference) and explain our choices of changes in accordance to these. Where changes to the text in the manuscript are made, the relevant paragraph is reproduced from the .pdf manuscript to this .docx response in *italic text*, with changes written in *italic green text*.

**Changes by authors**

Based on the feedback from the research community that we have received since the manuscript submission, we have made some minor changes to the manuscript. We hope that the reviewers share our notion that these modifications are improving the manuscript without changing its content.

In the following, we highlight the changes to the first submitted version of the manuscript.

Figure 1
Following the journal guidelines, we have added legends to Figs. 1b and 1c.

[Figure]

**Figure 1:** *Overview of the (                          )                ASCAL region. (a) Tracks of the icebreaker Polarstern during PASCAL (blue) and previous Arctic ship-based campaigns (orange to light green; see Sect. 5.4 for description). For the former, dark and bright colors indicate ocean-cruising (PSo; May 30 – June 5 and June 17–18, 2017) and ice-attached (PSi; June 6–16, 2017) positions, respectively. (b) Tracks of the aircraft Polar 5 (green) and Polar 6 (red) flights during ACLOUD May 23 – June 26, 2017, with later dates in brighter colors. (c) Track of PSo cruise and PSi position (blue). In (b) and (c), codes represent Longyearbyen (LYR), Ny-Ålesund (NYA), PSo entering the ACLOUD/PASCAL region, and PSi, while the shading and the dashed line represent the average sea ice concentration over the ACLOUD/PASCAL measurement period May 23 – June 26, 2017, and edge (defined by 15 % concentration) June 1979–2017, respectively (see Sect. 2.3 for data explanation).*

Figure 4 (previous Fig. 3)
Following the journal guidelines, we have added a legend for the atmospheric boundary layer (ABL) height to the lower panels in Fig. 4. Moreover, for consistency with the notation in the rest of the manuscript and in the updated Fig. 2a, we have changed the legend for wind barbs in the lower panels in Fig. 4. Finally, for improved readability, we have increased the contour lines in the upper panels in Fig. 4 and plotted less frequent arrows in the lower panels in Fig. 4.

[Figure]

**Figure 4:** *Vertical profiles of [left column] temperature and [right column] specific humidity measured at [upper row] Ny-Ålesund and [lower row] Polarstern over the ACLOUD/PASCAL measurement period May 23 – June 26, 2017. Blue circles indicate the height of the atmospheric boundary layer (ABL). Black contour lines in upper row panels represent the respective 1993–2016 average, while black arrows in lower row panels represent 2017 values of wind speed and direction. Dashed vertical lines distinguish the Polarstern ocean-crossing periods from the ice-attached period (June 6–16).*

Figure 5 (previous Fig. 4)
Following the journal guidelines, we have added a legend for the atmospheric boundary layer (ABL) height, the surface-based (SB) inversion, and the lifted (L) inversion to each panel in Fig. 5.

[Figure]

**Figure 5:** *Vertical profiles of (a) temperature and (b) specific humidity measured at Polarstern over the PASCAL measurement period May 28 – June 18, 2017. Pink and brown vertical lines indicate the vertical extent of the lowermost surface-based (SB) and lifted (L) inversions, respectively, while black contour lines indicate the atmospheric boundary layer (ABL) height corresponding to the blue circles in Fig. 3. Dashed vertical lines distinguish the Polarstern ocean-crossing periods from the ice-attached period (June 6–16).*

Figure 9 (previous Fig. 8)
To improve readability, in line with the journal guidelines, and for consistency with the other plots, we have re-plotted Fig. 8 with larger figure label sizes and made minor changes to the figure title and color bar labels.

[Figure]

**Figure 9:** *Climatologies (1979–2016; contours) and anomalies relative to climatologies (2017 minus 1979–2016; shading) of 700 hPa [left panels] geopotential height with key period median horizontal wind (vectors) and [right panels] virtual potential temperature for each ACLOUD/PASCAL key period based on ERA-I data. Key periods are defined as [first row] cold period (CP), [second row] warm period (WP), and [third row] normal period (NP).*

Figure 11 (previous Fig. 10)
Unfortunately, the statistical analysis of the cloud distribution from ACLOUD/PASCAL had to be reconducted. By mistake, we did not include all available IASI observations during this

period. Even so, even after including all available observations and reprocessing the data, the results did not change significantly, and the general conclusions drawn in this section were not affected. These small changes are now adopted in the graphics and text. Furthermore, we corrected a mistake in the plotting of the data where we overlaid the data over land by filled continents.

[Figure]

**Figure 11:** *Average [left column] cloud cover fractions and [right column] cloud top pressures for each ACLOUD/PASCAL key period from IASI. Box plots represent averages over the central ACLOUD/PASCAL region (76–82° N, 0–20° E; black boxes in map panels), with ticks indicating the 5 and 95 percentiles. The key periods are defined as [first row] the cold period (CP), [second row] the warm period (WP), and [third row] the normal period (NP).*

As a result, the 4th and 5th paragraphs in Sect. 4.4 now read:

*The high cloud cover during CP (Fig. 11a) was dominated by low-level clouds in the ACLOUD/PASCAL region with a median of the cloud top pressure around 770 hPa (Fig. 11b). However, this median would have risen to 790 hPa (corresponding to an altitude of about 2 km) by the exclusion of the last CP day (May 29; not shown), typical for the MCAO discussed in Sects. 3.1 to 3.3. This cloud regime is also well in alignment with the reduced 700 hPa geopotential height and virtual potential temperature in Figs. 9a and 9b, indicating that the region was dominated by a northerly flow (cf. Fig. 7a). Subsequently, low-level clouds developed over the open ocean and the cloud top longwave cooling led to a temperature inversion above the cloud (cf. Fig. 8a).*

*According to the time series of daily cloud cover fraction and top pressure (Fig. A3), the first six days of CP can clearly be classified as a stratus regime, which Eastman and Warren (2010) found to account for the majority of Arctic clouds in the May and June climatology. On the seventh and last day of CP (May 29), the change into another circulation regime is seen as the occurrence of high-level clouds (up to 300 hPa) increases in the central ACLOUD/PASCAL region due to their influence in its northern parts. Hence, no significant changes are observed near the surface (Fig. 2b).*

Furthermore, the 7th and 8th paragraphs in Sect. 4.4 now read:

*The lower cloud cover fraction during WP is associated with a change in cloud type, as cloud top pressure values were more than 100 hPa lower than in CP (Fig. 11d compared to Fig. 11b), highlighting the highest clouds observed during ACLOUD/PASCAL. A value of 650 hPa is typical for mid-level clouds, but can also result from a mixture of high- and low-level clouds. Average cloud top pressure values were also more homogeneous over the Nordic Seas in WP compared to CP. Clouds were then likely associated with synoptic disturbances, which brought moister air masses from both westerly and easterly directions (cf. Fig. 7b).*

*The cloud cover fraction and cloud top pressure in NP were in between those of CP and WP, with averages of about 80 % (Fig. 11e) and 700 hPa (Fig. 11f), respectively, but with larger spread in cloud top pressure. The strong variability was also observed on a day-to-day basis (Fig. A3), which was caused by a mix of low-, mid-, and high-level clouds. During this period, the air flow was dominantly northwesterly, and the proportion of low-level clouds increased with respect to WP.*

**Response to the review by Lana Cohen**

Concluding points
It seems the conclusions could be stated more explicitly in order to express clearly what the conclusions from this analysis are and why they are relevant. For example, this work shows that 1) the synoptic variability in this region drives much of the surface meteorology, atmospheric profiles, and cloud distribution; 2) the large-scale atmospheric circulation drives sea ice distribution during this period; 3) Ny Ålesund is fairly representative of the mid-to upper troposphere in this region; 4) understanding the boundary layer variability is important for understanding surface observations; 5) synoptic variability is linked to the large-scale atmospheric variability. These are just a few key results that are not included in the conclusions.

Based on your suggestion, we have added a new 4th paragraph to Sect. 6 that reads:
*The work presented in this paper shows that the synoptic variability in this region and time period is found to largely determine the surface meteorology, atmospheric profiles, and the cloud distribution. This synoptic variability is connected to the large-scale atmospheric variability, which itself was strongly linked to the sea ice distribution during the ACLOUD/PASCAL period. The analysis confirmed the conclusion by Kayser et al. (2017), who suggested that observations above Ny-Ålesund are fairly representative for the middle to upper troposphere in the ACLOUD/PASCAL region. However, for understanding surface observations, the knowledge of the boundary layer variability is key.*

Classical near-surface observations (P1 L5)
Not sure what is meant by "classical" near-surface... observations.

Based on your suggestion, the abstract now reads:
*The two concerted field campaigns Arctic CLoud Observations Using airborne measurements during polar Day (ACLOUD) and the Physical feedbacks of Arctic planetary boundary level Sea ice, Cloud and AerosoL (PASCAL) took place near Svalbard from May 23 to June 26, 2017. They were focused on studying Arctic mixed-phase clouds and involved observations from two airplanes (ACLOUD), an icebreaker (PASCAL), a tethered balloon, as well as ground-based stations. Here, we present the synoptic development during the 35 day period of the campaigns, using near-surface and upper-air meteorological observations, as well as operational satellite, analysis, and reanalysis data. Over the campaign period, short-term synoptic variability was substantial, dominating over the seasonal cycle. During the first campaign week, cold and dry Arctic air from the north persisted, with a distinct but seasonally unusual cold air outbreak. Cloudy conditions with mostly low-level clouds prevailed. The subsequent two weeks were characterized by warm and moist maritime air from the south and east, which included two events of warm air advection. These synoptical disturbances caused lower cloud cover fractions and higher-reaching cloud systems. In the final two weeks, adiabatically warmed air from the west dominated, with cloud properties strongly varying in between the range of the two other periods. Results presented here provide synoptic information needed to analyze and interpret data of upcoming studies from ACLOUD/PASCAL, while also offering unprecedented measurements in a sparsely observed region.*

In other words, we have removed the word 'classical' before 'near-surface and upper-air meteorological observations'.

Radiosonde measurements specification (P4 L23 & P5 L1)
Include "Vaisala" before RS41 and RS92

Based on your suggestion, we have added "Vaisala" before RS41 and RS 92 in the 2nd and 4th paragraphs in Sect. 2.2, which now read:
*The AWIPEV research base in Ny-Ålesund is located about 100 km northwest of Longyearbyen. Since 1992, AWI routinely operates a variety of atmospheric measurements in Ny-Ålesund, which were intensified during ACLOUD/PASCAL. The frequency of the daily radiosonde measurements was increased to four* Vaisala *RS41 launches per day, providing 6-hourly vertical profiles of temperature, humidity, pressure, and wind speed and direction with about 5 m vertical resolution (Maturilli, 2017a,b).* By integration, 6-hourly integrated water vapor (IWV) is retrieved. Standard atmospheric *parameters were observed every minute at the surface (Maturilli et al., 2013), of which surface pressure and 2 m temperature are presented here.*
*Four daily* Vaisala *RS92 radiosondes were launched from Polarstern during most of* the *ACLOUD/PASCAL* period *(Schmithüsen, 2017). These retrieved vertical profiles are compared to the Ny-Ålesund data,* as are every minute pressure observations at 16 m height and temperature observations at 29 m height on-board Polarstern (Schmithüsen, 2018). Detailed *information of the* instrumentation of *Polarstern* during PASCAL are summarized by *Macke and Flores (2018) and Wendisch et al. (2018).*

Polarstern data explanation (P5 L1-3)
These first two sentences are poorly worded. Please clarify.

Based on your suggestion, we have rewritten the two relevant sentences in the 4th paragraph in Sect. 2.2, which now reads:
*Four daily* Vaisala *RS92 radiosondes were launched from Polarstern during most of* the *ACLOUD/PASCAL* period *(Schmithüsen, 2017). These retrieved vertical profiles are compared to the Ny-Ålesund data,* as are every minute pressure observations at 16 m height and temperature observations at 29 m height on-board Polarstern (Schmithüsen, 2018). *For more information on the dedicated Polarstern campaign instruments, please see Macke and Flores (2018) and Wendisch et al. (2018).*

IASI technical details (P5 L34)
Explain what "level 2 cloud cover" is.

Based on your suggestion, the 5th paragraph in Sect. 2.3 now reads:
*IASI is part of the MetOp series of polar orbiting satellites and has a swath width of about 2,200 km (EUMETSAT, 2017). Due to the meridional convergence of the orbits the temporal sampling of the ACLOUD/PASCAL region is high, with several overpasses per day.* Here, we use cloud cover fraction and cloud top pressure products (level 2, version 6) retrieved from IASI radiance measurements *to investigate the distribution of clouds. Cloud detection is performed followed by a retrieval of cloud top pressure using the $CO_2$-slicing technique for*

*each IASI field of view (e.g., Lavanant et al., 2011). As shown by Lavanant et al. (2011), the retrieval of cloud top pressure works best for homogeneous, opaque clouds common for Arctic regions and is difficult in broken and multi-layer cloud situations.*

Cloud retrieval scheme explanation (P6 L2-5)
Sentence is poorly worded.

Please see IASI technical details (P5 L34) above.

Ny-Ålesund radiosonde assimilation into ERA-I (P6 L14)
It should be noted that the Ny Ålesund radiosondes are assimilated into ERA-Interim. Were the extra sondes launched during this campaign also assimilated?

Based on your suggestion, we have added information on Ny-Ålesund and Polarstern radiosonde assimilation to the 2$^{nd}$ paragraph in Sect. 2.4, which now reads:

> *The European Re-Analysis Interim (ERA-I; Dee et al., 2011) provided data of atmospheric circulation, temperature, and humidity for the ACLOUD/PASCAL region. This reanalysis provides the best description of the state of the atmosphere by assimilating a wealth of observations, including satellites, radiosondes (also the ones described in Sect. 2.2 from Ny-Ålesund and Polarstern), and land stations, and is found to be well-suited for the northern regions (Jakobson et al., 2012; Chung et al., 2013; Lindsay et al., 2014).*

Synoptic description (P7 L2 + P7 L3-7 + P7 L8 + P7 L10 + P7 L13)
Describe the synoptic situation that is driving this cold air outbreak. This could be shown using the synoptic maps that are included in Fig A1.

This paragraph is not at all clear. Please rework to clarify.

"The variable wind direction over Svalbard finally caused the highest precipitation…" It cannot be said that wind direction caused precipitation, please explain this more clearly.

Based on your suggestion and in line with Sect. 2.4, we have significantly modified Sect. 3.1 in the updated manuscript. Specifically, we have re-plotted Fig. A1 with ECMWF analysis data instead of Wetterzentrale data, removed information on 500 hPa temperature from it, replaced its panel for May 29 with one for May 27, moved it from the appendix to Sect. 3.1, and renamed it as Fig. 3. There, we have integrated the figure stronger and improved the synoptic description. As we see it, this addresses all your points above, which you rightfully pointed out being sources of improvements for the manuscript.

[Figure]

**(a)** *May 26, 2017*  **(b)** *May 27, 2017*

**(c)** *June 2, 2017*  **(d)** *June 6, 2017*

500 508 516 524 532 540 548 556 564 572
Geopotential height (gpdm)

[revised manuscript text omitted]

While acknowledging that highlighting the periods discussed in Sect. 3.1 in Fig. 2 could be helpful, we prefer to avoid this here. The reasons are that A) Fig. 2 already is busy with information as it is now, meaning that added information would – as we see it – only confuse the reader; B) the periods discussed in Sect. 3.1 are partly overlapping, meaning that indication of these periods would need to overlap in Fig. 2 too, also being a possible source of confusion; and C) the periods discussed in Sect. 3.1 should not be confused with the key periods discussed in Sect. 4, as first indicated in Fig. 5a.

Atmospheric river description (P7 L18)

How do you know there was an atmospheric river? It is not apparent in Fig A1c that it would it be coming from the east. This should be clarified.

Based on your review, we gratefully realized that the reference to the atmospheric river reaching the ACLOUD/PASCAL region around May 29 was misplaced. Hence, in the updated manuscript, this clause is taken out of the 5th paragraph in Sect. 3.1 and instead added to the last line in the 4th paragraph, which now reads:

*The following days saw IWV and temperature substantially increase in Ny-Ålesund, from 6 kg m⁻² on May 28 to 14 kg m⁻² on May 30 (Fig. 2c) and from -10° C on May 29 to +7° C on May 31 (Fig. 2b), respectively. The former increase was related to a narrow band of high IWV and intense integrated water vapor transport (IVT), identified as an atmospheric river, which reached Svalbard from western Siberia on May 30 (Fig. A1a). In this period, precipitation occurred in the ACLOUD/PASCAL region but was confined to a small area. After this event, the wind direction turned northerly again due to a strong low that formed southeast of Svalbard (not shown), advecting more cold air from the ice-covered areas.*

This statement follows from an additional analysis on atmospheric rivers, which is now included as Fig. A1. This was conducted by the two new co-authors, I. V. Gorodetskaya and C. Viceto from the University of Aveiro. In the process, they also discovered a new event on June 6, which together with June 9 and 13 are depicted in Fig. A1.

[Figure]

**Figure A1:** *Vertically integrated water vapor (IWV; in kg m⁻²; shading), 700 hPa geopotential height (in m; white contours), and sea ice edge (defined by 15 % concentration; grey contours) for (a) 06 UTC on May 30, (b) 12 UTC on June 6, (c) 12 UTC on June 9, and (d) 00 UTC on June 13, 2017, based on ERA-I data. In (a) and (b), red arrows indicate the IWV transport (IVT; in kg m⁻¹ s⁻¹) within the atmospheric rivers affecting Ny-Ålesund.*

As a result, the 5th paragraph in Sect. 1 now reads:

*The Arctic North Atlantic sector is particularly different as compared to other Arctic regions. It is frequently affected by cyclones associated with the Icelandic low (Serreze et al., 1997), which transport heat and moisture into the Arctic, driving the transitions between radiatively clear and cloudy states (Stramler et al., 2011; Graham et al., 2017). It is also the region of most frequent intrusions of moist and warm air entering the Arctic (Woods and Caballero, 2016; Nash et al., 2018), which affects the marginal ice zone (MIZ) as well as the atmospheric thermodynamic structure, and the formation, distribution, and properties of clouds (Johansson et al., 2017). In this area, the conditions are favorable for studying the coupling of the ABL clouds with cyclones and large-scale circulation. Furthermore, the proximity to the sea ice edge north of Svalbard allows an investigation of the cloud microphysical changes during air mass transformations during both moist air intrusions and cold air outbreaks (Young et al., 2016). Overall, the area close to Svalbard enables studies of the response of cloud properties to changes in local sea ice conditions, surface heat and moisture fluxes, in the thermodynamic structure of the lower atmosphere, and to the large-scale synoptical conditions that control the origin of the air mass in which the clouds form.*

Similarly, the 3rd paragraph in Sect. 2.4 now reads:

*ERA-I data were acquired on a 0.75° x 0.75° horizontal grid for the period of May–June 1979–2017. These data served as the basis for the identification of atmospheric rivers affecting Ny-Ålesund discussed in Sect. 3.1, following from the algorithm by Gorodetskaya et al. (2014) and adapted for the Arctic. In the calculation of the weather events in Sects. 3.3 and 5.3, 6-hourly 850 hPa and skin temperature and 850 hPa geopotential were used. Parameters presented in Sect. 4.2 are based on 6-hourly 700 hPa geopotential, zonal and meridional winds, temperature, and specific humidity. 700 hPa virtual potential temperature is estimated from the last two and is therefore a merged measure of air temperature and humidity (Etling, 2008). Daily 1000 hPa geopotential was obtained for Sect. 5.1.*

Furthermore, the 7th and 9th paragraphs in Sect. 3.1 now read:

*The period of warm temperatures at begin of June represent the highest positive temperature anomaly recorded during ACLOUD/PASCAL. In Ny-Ålesund, 7° C and 8° C were observed on May 31 and June 6, respectively (Fig. 2b), both being indications of warm air advections (WAAs; Tjernström et al., 2015). The latter event was accompanied by an increase of IWV to 15 kg m⁻² (Fig. 2c), which was linked to another atmospheric river episode reaching Ny-Ålesund from the east (Fig. A1b).*

*Both Ny-Ålesund and Polarstern experienced distinct drops in near-surface pressure associated with increases of near-surface air temperature and IWV around June 13 (Figs. 2a to 2c). The air mass reaching the ACLOUD/PASCAL region on this day had an European origin but circled once around Svalbard before arriving Ny-Ålesund from the north (shown later in Fig. 7c). The peaks in the IWV observed in Ny-Ålesund on June 9 and 13 can be explained by air*

*masses with high IWV but no intense IVT on those days (Figs. A1c and A1d). For the remainder of the measurement period, surface pressure, near-surface air temperature, and IWV observed in Ny-Ålesund were close to the long-term average, as well as close to Polarstern values until the icebreaker left the ice (June 18).*

Snow melt season discussion (P7 L26)

This paragraph is not clear. This first mentions beginning of snow melt season, but then switches to first snow-free day and albedo. Please make this more clear.

Based on your suggestion, we have rewritten the 6th paragraph in Sect. 3.1, which now reads:

*The 17° C warming within only two days in Ny-Ålesund, which marked the beginning of the snow melt season on May 29 (Fig. 2b), was also imprinted in the time series of snow albedo obtained by the surface radiation measurements. From this date, the surface albedo temporarily decreased from 0.9 to lower values, before it rapidly dropped to below 0.1 by June 14, when the snow had completely disappeared. This development agrees with the climatology of Ny-Ålesund, which reports the first snow-free day between May 30 and July 5 since the beginning of the BSRN measurements in late 1992.*

Lifted inversions discussion (P9 L14-31)

The discussion of lifted inversions does not add much to the overall analysis. Perhaps it could just be mentioned briefly or merged into the discussion of Figure 3?

Figure 5 was one of the figures mentioned above that in itself is not necessarily crucial for the scientific story of this manuscript, but will be of high importance to upcoming papers from ACLOUD/PASCAL, especially those focusing on the ABL. Originally, we had all the information from Figs. 4 (previous 3) and 5 in one figure, but found this to be to make too busy panels and thus be hard to interpret. For these reasons, as well as for the different x- and y-axis compared to Fig. 4, we prefer keeping Figs. 4 and 5 as separate figures.

That being said, we agree with you in that the discussion of Fig. 5 could be more concise. Hence, in the updated manuscript, the last four paragraphs in Sect. 3.2 in the submitted manuscript are condensed into two paragraphs that read:

*The radiosondes, given their high vertical resolution, further allow the investigation of temperature and humidity inversion variabilities during the ACLOUD/PASCAL period. Inversions are a dominant feature of the Arctic wintertime boundary layer. In spring, the frequency of inversions decreases but still significantly impacts the atmospheric temperature, moisture, and energy exchange. Temperature inversions have significant impacts on the atmospheric stratification (Lesins et al., 2010) and manipulate the vertical distribution of long-wave radiation (Bintanja et al., 2011). In particular, specific humidity inversions are known to be a source of long-wave radiative heating of the surface during cloud-free conditions (Devasthale et al., 2011) and are relevant for cloud physics (Sedlar et al., 2012; Solomon et al., 2011). For these reasons, Fig. 5 provides a more detailed picture of the boundary layer processes during ACLOUD/PASCAL, showing the retrieved altitudes of surface-based and lifted inversions observed in the radio soundings over Polarstern. The inversions*

*were identified following the methods described in Andreas et al. (2000), Kahl (1990), and Kayser et al. (2017).*

*During the ACLOUD/PASCAL measurement period, inversions were found in most soundings for both temperature and specific humidity, particularly throughout June when Polarstern was located in areas covered by sea ice. During the period of the ice floe camp (June 6–16), an enhanced occurrence of surface-based inversions was found. This was caused by temperature and humidity advections above the boundary layer, while the ice surface remained at a temperature of 0° C, stabilized by the snowmelt. In general, a lifted temperature inversion was present when the ABL was relatively high (up to 700 m), while a surface-based temperature inversion was observed when the ABL was shallow (about 200 m).*

CWT analysis and discussion (P10 L27-30)
It doesn't seem that the CWTs add much to the discussion or understanding of the synoptics or large scale circulations. In fact it only raises questions (eg. CWTs for Ny Ålesund and Polarstern are more different that one might expect!). I think this discussion (and Fig 5b) could be left out altogether.

Based on your suggestion, Fig. 6b (previous Fig. 5b) and the last four paragraphs in Sect. 3.3 in the submitted manuscript are deleted, while the 4[th] paragraph in Sect. 3.3 now reads:

*As shown in Fig. 6, the MCAO index varied considerably over the first three weeks of ACLOUD/PASCAL. During the first eight days (May 23–30), values were above the median of the climatology and mostly exceeded the 95 % percentile until May 28. Corresponding to the anomalously cold and dry air observed in Figs. 2 and 4, we identify a MCAO event during the first week of the measurement period (maximum May 23 in Fig. 6). The MCAO index then dropped significantly from +2 K on May 28 to -11 K on May 31, remaining below the median until June 15. During these two weeks, values remained below -12 K (i.e., below the 25 % percentile) except for June 7. In combination with the temperature and humidity time series (Figs. 2 and 4), we identify two WAA events during the second and third week of the measurement period (minima June 5 and 10 in Fig. 6). After June 14, the MCAO index increased again and leveled around the long-term median between -5 K and -7 K, indicating normal weakly unstable conditions in the lower troposphere (i.e., neither MCAO nor WAA conditions).*

Consequently, we have replaced references to Fig. 6b with references to Figs. 7 (previous 6) and 9 and Video S1. As a result, the 4[th] and 7[th] paragraphs in Sect. 4.4 now read:

*The high cloud cover during CP (Fig. 11a) was dominated by low-level clouds in the ACLOUD/PASCAL region with a median of the cloud top pressure around 770 hPa (Fig. 11b). However, this median would have risen to 790 hPa (corresponding to an altitude of about 2 km) by the exclusion of the last CP day (May 29; not shown), typical for the MCAO discussed in Sects. 3.1 to 3.3. This cloud regime is also well in alignment with the reduced 700 hPa geopotential height and virtual potential temperature in Figs. 9a and 9b, indicating that the region was dominated by a northerly flow (cf. Fig. 7a). Subsequently, low-level clouds developed over the open ocean and the cloud top longwave cooling led to a temperature inversion above the cloud (cf. Fig. 8a).*

*The lower cloud cover fraction during WP is associated with a change in cloud type, as cloud top pressure values were more than 100 hPa lower than in CP (Fig. 11d compared to Fig. 11b), highlighting the highest clouds observed during ACLOUD/PASCAL. A value of 650 hPa is*

*typical for mid-level clouds, but can also result from a mixture of high- and low-level clouds. Average cloud top pressure values were also more homogeneous over the Nordic Seas in WP compared to CP. Clouds were then likely associated with synoptic disturbances, which brought moister air masses from both westerly and easterly directions (cf. Fig. 7b).*

Moreover, the 5th and last paragraphs in Sect. 5.4 (previous Sect. 5.3) now read:

*ASCOS was dominated by anticyclonic atmospheric circulation, while cyclonic circulation prevailed during AOE-96 and AOE-2001 (Tjernström et al., 2012). During the ACLOUD/PASCAL measurement period, we found strong daily variability (Video S1), with cyclonic and anticyclonic circulation governing CP and WP, respectively (Figs. 9a and 9c). Nevertheless, similar to ACLOUD/PASCAL (Fig. 7), significant differences in air flow regimes were also observed during ASCOS (Fig. 9 in Tjernström et al., 2012).*

*In general, ACLOUD/PASCAL was to a low degree influenced by synoptic cyclones, as indicated by the few significant changes in the temperature and humidity time series (Figs. 2b and 2c) in association with the changes in the pressure time series (Fig. 2a). In this respect, the conditions during N-ICE2015 were different, when a persistent and anomalous low pressure centered over the Barents Sea dominated the corresponding season (Cohen et al., 2017). In 2015, this caused more abrupt shifts in cloud cover due to the associated cyclonic circulation (Cohen et al., 2017; Graham et al., 2017; Kayser et al., 2017); in 2017, we observed the cloudiest conditions in association with cyclonic circulation (Figs. 9a and 11a). We also found no significant precipitation events to follow from pressure drops (Figs. 2a and 2c here compared to Fig. 3a in Cohen et al., 2017).*

Furthermore, the 2nd paragraph in Sect. 6 now reads:

*Time series of the data from Ny-Ålesund (at 79° N, 12° E) and Polarstern ocean-crossing (in the Nordic Seas north of the Arctic Circle) and ice-attached locations (at about 82° N, 10° E) during the 35 day measurement period are presented and compared to the long-term near-surface and radiosonde measurements conducted in Ny-Ålesund. Additionally, we computed the marine cold air oubreak (MCAO) index and compared this to its climatology of the region.*

Finally, the acknowledgements now read:

*We gratefully acknowledge the support from the Transregional Collaborative Research Center (TR 172) "ArctiC Amplification: Climate Relevant Atmospheric and SurfaCe Processes, and Feedback Mechanisms (AC)³", which is funded by the German Research Foundation (Deutsche Forschungsgemeinschaft; DFG). We thank D.C. Strack for providing data used in Fig. 10, as well as M. Gerken and C. Patiliea for preparing plots used in Figs. 10 and 13, respectively. C. Melsheimer provided the algorithm used in Videos S1 and S2. M. Kayser should also be mentioned for valuable comments on an earlier version of this manuscript. M. Rautenhaus is acknowledged for providing the Mission Support System (MSS; Rautenhaus et al., 2012) for flight planning during ACLOUD, as well as J. Ungermann and R. Bauer for technical support.*

In other words, we have replaced the reference to Fig. 6a (previous Fig. 5a) with Fig. 6 (previous Fig. 5) and removed the mentioning of Fig. 6b.

Backward air mass trajectory analysis and discussion (P11-12)
It is not clear that the back trajectories are adding much to the analysis, and in this section there are some issues with these trajectories that are not addressed. The main issues are 1) the very large uncertainties that exist for going back 10 days (only 5 days is often used for this reason) should be addressed; 2) spatially averaging trajectories with large variability can be very misleading and this should be acknowledged; and 3) it seems overly complicated to use PES instead of a simple residence time (perhaps you could use HYSPLIT instead of FLEXPART)?

We acknowledge that the background explanation of FLEXPART was rather short in the submitted manuscript. For this reason, we have extended the description of the model in the manuscript and also included some further information below.

Figure 7 was another figure asked for by the research community. While having some overlap with Fig. 9, we argue that its inclusion (with its daily trajectories separated into three key period panels) offers a smooth transition from the time series in Sect. 3 to the key periods in Sect. 4, while also being visually easy to interpret for the reader and thus clarify the distinction between the key periods early in Sect. 4.

Furthermore, the FLEXPART model significantly helped us to confirm the results based on the measurements from Ny-Ålesund in Sect. 3, both with respect to the change in the weather regimes and with respect to the identification of the three different key synoptic periods and their corresponding air mass history. From the FLEXPART simulations, we could confirm that there were distinct differences in the transport pathways in the days prior to the measurements. For example, it becomes clear that the cold period (CP) is linked to air masses that resided most of the time prior to the measurements in the high Arctic (> 65° N) and that the warm (WP) and neutral (NP) periods are much more influenced by air masses from midlatitudes (< 65° N). This is not obvious from the single point measurements in Ny-Ålesund presented in Sect. 3.

Using FLEXPART also allowed us to address the underlying uncertainty more comprehensively than using a simple trajectory model like HYSPLIT. FLEXPART provides the possibility to calculate the fate of many individual particles (equal to individual trajectories) simultaneously. By using a large ensemble of particles in each simulation, the statistical confidence in the results increases. In FLEXPART, the atmospheric transportation of these particles is not only relying on the average wind, as in common trajectory models, but also on subgrid-scale turbulence and convection (see Stohl et al., 2005, for more information). In each simulation (i.e., each day), we released more than 400.000 tracer particles (which is equal to trajectories). Also using these large ensembles of tracer particles allow us to perform longer calculations than classical kinematic trajectory studies. Hirdman et al. (2010) even used 20 days backward trajectories to study pollution events in the Arctic. Other studies also rely on ten days trajectories (e.g., Collins et al., 2017). Even in midlatitude studies, ten days or more were used to study air mass histories, even though the dynamics often are much more influenced by strong shear, deformation, and divergence (e.g., Eckhardt et al., 2003).

These strengths of the FLEXPART simulations are included in our Fig. 7, which not only shows PES but also the center of mass trajectories. These trajectories are calculated based on the individual particle positions. Compared to classical kinematic trajectories, they are also affected by sub-grid turbulence and unresolved convection. That being said, these center of mass trajectories are closer to the kinematic trajectories than PES output. Comparing the center of mass trajectories between the three time periods does not show such a clear difference because almost all center of mass trajectories either point north- or eastward. However, PES values clearly distinguish CP and WP, with high values over the central Arctic Ocean in the former and values above zero at midlatitudes (< 65° N) in the latter.

PES values result from FLEXPART calculations in backward mode. This emission sensitivity function is directly related to the residence time of a particle in a model grid box. It also measures the simulated concentration at the receptor that a source of a unit strength in this model box would produce for an inert tracer not affected by any removal process. For more clarification on PES, we refer to Stohl et al. (2005), Seibert and Frank (2004), and Hirdman et al. (2010).

Using PES, one of our objectives was to show that the three key periods were not only different in terms of meteorological conditions in Ny-Ålesund but also in terms of air mass history. In doing so, we wanted to highlight the various overall distributions of the three key periods and not the individual days composing these key periods themselves, even if this meant that we might have lost some information on the individual days through temporal averaging.

Relevant references are:
- Collins, D., Burkart, J., Chang, R., Lizotte, M., Boivin-Rioux, A., Blais, M., Mungall, E., Boyer, M., Irish, V., Massé, G., Kunkel, D., Tremblay, J.-É., Papakyriakou, T., Bertram, A., Bozem, H., Gosselin, M., Levasseur, M., and Abbatt, J.: Frequent ultrafine particle formation and growth in Canadian Arctic marine and coastal environments, Atmos. Chem. Phys., 17, 13119–13138, https://doi.org/10.5194/acp-17-13119-2017, 2017.
- Eckhardt, S., Stohl, A., Beirle, S., Spichtinger, N., James, P., Forster, C., Junker, C., Wagner, T., Platt, U., and Jennings, S.: The North Atlantic Oscillation controls air pollution transport to the Arctic, Atmos. Chem. Phys., 3, 1769–1778, https://doi.org/10.5194/acp-3-1769-2003, 2003.
- Hirdman, D., Sodemann, H., Eckhardt, S., Burkhart, J., Jefferson, A., Mefford, T., Quinn, P., Sharma, S., Ström, J., and Stohl, A.: Source identification of short-lived air pollutants in the Arctic using statistical analysis of measurement data and particle dispersion model output, Atmos. Chem. Phys., 10, 669–693, https://doi.org/10.5194/acp-10-669-2010, 2010.
- Seibert, P. and Frank, A.: Source-receptor matrix calculation with a Lagrangian particle dispersion model in backward mode, Atmos. Chem. Phys., 4, 51–63, https://doi.org/10.5194/acp-4-51-2004, 2004.

- Stohl, A., Forster, C., Frank, A., Seibert, P., and Wotawa, G.: Technical note: The Lagrangian particle dispersion model FLEXPART version 6.2, Atmos. Chem. Phys., 5, 2461–2474, https://doi.org/10.5194/acp-5-2461-2005, 2005.

In the manuscript, two paragraphs are added to Sect. 2.4, which read:

*Using FLEXPART, we continuously released 480.000 individual air parcels close to the surface at the location of Ny-Ålesund for every day of the campaign period. These air parcels represent an inert air mass tracer and were further traced back in time for another ten days. The distribution of this air mass tracer – and thus the pathway of the trajectories through the atmosphere – does not only depend on the mean wind given in the operational analysis data but also on turbulent motions (Stohl et al., 2005). These motions also affect the center of mass trajectories, contrasting the commonly used kinematic trajectories that only depend on the mean wind field from meteorological input data. Using this amount of individual air parcels and considering the turbulent motions allow us to obtain a better estimate of the distribution of the air masses, which potentially affected the observations in Ny-Ålesund.*

*In backward mode, FLEXPART provides potential emission sensitivity (PES), which is the response function of the source-receptor matrix (Seibert and Frank, 2004). PES is directly related to the residence time of a particle in a model grid box and measures the simulated concentration at the receptor that a source of a unit strength in this model box would produce for an inert tracer not affected by any removal process (see also Stohl et al., 2005; Hirdman et al., 2010). We used PES available on a 0.25° grid in the horizontal, which represents the entire tropospheric column.*

With most of their information moved to Sect. 2.4, the two first paragraphs in Sect. 4.1 are merged and now read:

*To assess differences in the air mass histories of the three key periods defined above we compare their mean trajectories. This analysis was performed using FLEXPART in backward mode, with input data from ECMWF operational analysis (cf. Sect. 2.4). In addition to the temporal means of PES over each key period, Fig. 7 also shows the daily center of mass trajectories of the respective key period.*

Figure 8 (previous Fig. 7) discussion (P12-13)
This section discussing Fig 7 could be much more concisely written. What are the key features of Fig 7? Rather than describing every feature of the plots, just focus on 2 or 3 important points.

Based on your suggestion, the last eight paragraphs in Sect. 4.1 in the submitted manuscript discussing Fig. 8 are condensed into four paragraphs that read:

*Figure 8 shows the varying profiles of temperature and specific humidity as observed over Ny-Ålesund and Polarstern during the three key periods. Only Ny-Ålesund data are included in Figs. 8a and 8b due to the southerly location of Polarstern during the first campaign week, unrepresentative of the Arctic. In Figs. 8c to 8f, Polarstern data are split in two profiles to differentiate its ice-attached and ocean-cruising locations (cf. Fig. 1c).*

*While in the end of May, the first key period (CP) was characterized by relatively cold and dry air above Ny-Ålesund, with temperatures continuously below 0° C and humidity mostly below 2 g kg$^{-1}$ (Figs. 8a and 8b). The nearly isothermal average profile between 900*

*hPa and 800 hPa is consistent with the top of the frequent low-level clouds observed during this period (shown later in Fig. 11b). No inversions prevail in the average temperature and humidity profiles, although some individual soundings show humidity inversions around 820 hPa, where the radiosondes escape the mountain ridges and enter the synoptic flow.*

*During the second key period (WP), two features were noteworthy. Firstly, above Ny-Ålesund, a rather weak temperature inversion (< 1° C) was detected in the average profile at 910 hPa, while the lower troposphere had warmed (+10° C) and moistened (+ 5 g kg$^{-1}$) substantially with respect to CP (Figs. 8c and 8d compared to Figs. 8a and 8b). Secondly, during WP above Polarstern, a marked temperature inversion of about 5° C prevailed in the lowest 100 hPa for both ice-attached and ocean-cruising periods. Moreover, elevated humidity inversions of 1.0–1.5 g kg$^{-1}$ were detected in individual soundings.*

*During the third key period (NP), the averaged temperature profile above Ny-Ålesund formed a similar shape as during CP, revealing no inversions but a warming of about 10° C (Fig. 8e) and a moistening of about 2 g kg$^{-1}$ (Fig. 8f). Above Polarstern, weak temperature inversions were present in the average profiles. Individual soundings with an elevated humidity inversion appeared at 900 hPa above both Ny-Ålesund and the ice-attached Polarstern. This feature was not seen above the ocean-cruising Polarstern, possibly due to the few soundings in this profile (two days only).*

Figures A3 and A4 deletion (P13 L26)
Figures A3 & A4 are not mentioned further in the text, and can therefore be eliminated.

Based on your suggestion, we have removed Figs. A3 and A4 from the appendix. However, as they were asked for by the research community, we have uploaded two corresponding videos of these figures as Videos S1 and S2 in the supplementary material in accordance with the journal guidelines, which then also improves their readability due to the small appearance of the 35 daily panels each in Figs. A3 and A4.

As a result, the 1st paragraph in Sect. 4.2 now reads:
*Figure 9 complements Figs. 7 and 8 by picturing the contrasting atmospheric circulation, temperature, and humidity of the three key periods based on ERA-I data. Here, Figs. 9a, 9c, and 9e illustrate the 700 hPa geopotential height and horizontal wind of CP, WP, and NP, respectively, while the relative temperature and humidity of these periods are depicted in Figs. 9b, 9d, and 9f. In addition to their climatology, each panel depicts the anomalous conditions of the three key periods compared to their respective climatology. A more detailed evolution of the atmospheric circulation and thermodynamics observed during ACLOUD/PASCAL is presented by daily fields of these measures in Videos S1 and S2 in the supplementary material. The 700 hPa level represents the main flight level during ACLOUD (Wendisch et al., 2018).*

Moreover, the acknowledgements now read:
*We gratefully acknowledge the support from the Transregional Collaborative Research Center (TR 172) "ArctiC Amplification: Climate Relevant Atmospheric and SurfaCe Processes, and Feedback Mechanisms (AC)³", which is funded by the German Research Foundation (Deutsche Forschungsgemeinschaft; DFG). We thank D.C. Strack for providing data used in Fig. 11, as well as M. Gerken and C. Patiliea for preparing plots used in Figs. 10 and 13,*

*respectively. C. Melsheimer provided the algorithm used in Videos S1 and S2. M. Kayser should also be mentioned for valuable comments on an earlier version of this manuscript. M. Rautenhaus is acknowledged for providing the Mission Support System (MSS; Rautenhaus et al., 2012) for flight planning during ACLOUD, as well as J. Ungermann and R. Bauer for technical support.*

Section 5.1 renaming (P16 L22)
Very minor point, but perhaps Section 5.1 should be titled "Large-scale indices" since it does not actually deal with atmospheric dynamics in any way.

Based on your suggestion, we have renamed Sect. 5.1 to *Large-scale circulation indices*.

Figure 14 (previous Fig. 13) deletion (P18 L20)
Not sure you need Figure 13; I think the text description here is enough.

While acknowledging your suggestion of leaving out Fig. 14, we argue that it is important for visualizing the description of the MCAO and WAA anomalies in Sect. 5.3 (previous Sect. 5.2). Moreover, by replacing the number of events in the figure by actual years as we have done in the updated manuscript (see replica below), we think that the inclusion of Fig. 14 becomes clearer as it will then add another dimension to the text.

[Figure]

**Figure 14**: *(a) Marine cold air outbreak (MCAO) and (b) warm air advection (WAA) durations and intensities for the eastern Greenland Sea (75.00–80.25° N, 4.50–10.50° E) over the ACLOUD/PASCAL comparison period May 23 – June 26, 1998–2017, based on ERA-I data. Colored boxes represent the number of MCAO and WAA events over 1998–2016, with specific years indicated in white. Black bullseyes represent 2017 events.*

SHEBA comparison (P19 L8 + P19 L14-18)
Make more explicit that SHEBA was in a completely different part of the Arctic and that there are considerable regional differences.
Not sure that any of this comparison to SHEBA is relevant due to different region and different climatology. I think it just confuses things, I would leave this out.

Based on your suggestions, the 2nd and 3rd paragraphs in Sect. 5.4 are merged and shortened to now read:

*SHEBA (cf. Fig. 1a) was the first field campaign to include a full year of Arctic measurements (Uttal et al., 2002). Taking place from October 1997 to October 1998, its main objective was to advance the understanding of the coupled ocean-ice-atmosphere processes in models. While taking place in the ice pack of the Beaufort Sea on the opposing side of the Arctic Ocean, some comparisons to ACLOUD/PASCAL can still be made. During May and June 1998, temperature inversion heights of about 200–700 m and persistent cloudiness (80–100 %) characterized the SHEBA ice camp (Uttal et al., 2002). Over the same months in 2017, we observed inversion heights both shallower (about 100 m) and deeper (about 1400 m) north of Svalbard (Fig. 5a), along with cloudy conditions in the whole region (Fig. 11). While there are considerable regional differences between the Beaufort Sea and the Fram Strait, the snow melt season began May 29 and ended during the first half of June both during SHEBA and ACLOUD/PASCAL (Fig. 2b).*

Moreover, to more explicitly state why we included Sect. 5.4, the 1st paragraph in Sect. 5.4 now reads:

*The few observations in the ACLOUD/PASCAL region partly explain the motivation for the field campaigns. Paradoxically, this also makes it hard to compare the data shown in this manuscript to other studies. Nevertheless, with differences in years, seasons, locations, and set-ups taken into account, such a comparison is still relevant for understanding the rapidly changing Arctic climate system. In this way, ACLOUD/PASCAL provides an important addition to earlier campaigns, as well as serving as a benchmark for upcoming Arctic field campaigns (e.g., the Multidisciplinary drifting Observatory for the Study of Arctic Climate; MOSAiC; IASC, 2016).*

As a result, also the last paragraph in Sect. 6 now reads:

*Our focus was limited to the North Atlantic sector of the Arctic. Hence, the results presented here do not necessarily translate to the entire Arctic climate system because the regional differences are too large (e.g., Serreze et al., 2011; Cavalieri and Parkinson, 2012; Koyama et al., 2017). For example, sea ice coverage in the region was anomalously high and reached far south as a result of the strong southward drift during CP and, albeit weaker, still southward drifts during WP and NP. Nevertheless, considering the sparsely observed Arctic region, the extensive ACLOUD/PASCAL campaign offers unique measurements covering the entire tropospheric column, with observations over the open ocean, sea ice, and snow. Most measurements performed during ACLOUD/PASCAL will be continued in the framework of MOSAiC, including a one-year ice drift of Polarstern and numerous aircraft- and ground-based activities. Thus, while MOSAiC will strongly benefit from the results and experiences gained from ACLOUD/PASCAL, the continuity of observations in this Arctic region is anticipated to considerably improve the understanding of the cloud-related processes in the Arctic*

*atmosphere, as well as the ocean-ice-atmosphere interaction from turbulent and radiative energy fluxes. Ultimately, this will strengthen synoptic forecasting in weather models, benefiting actors beyond the scientific community.*

SHEBA and TARA comparisons (P19 L19-24)
This paragraph is very confusing, since both Tara and SHEBA are mentioned in comparison to ACLOUD/PASCAL. Please make this more clear.

Based on your suggestion, we have rewritten the 4th paragraph in Sect. 5.4, which now reads:
*The drifting ice station TARA took place in the central Arctic Ocean during the International Polar Year September 2006 to September 2007 (cf. Fig. 1a) and thus within the trend of rapidly rising Arctic temperatures (Vihma et al., 2008). Even so, the summer (as defined by snow/sea ice temperature) started later at TARA than at SHEBA nine years earlier: on June 9 compared to May 30. Similarly, the mean profiles from April to August were warmer and moister during SHEBA (Vihma et al., 2008). These warmer conditions might be a result of the more northerly location of Tara compared to SHEBA. While also taking place mostly north of SHEBA, mean profiles during ACLOUD/PASCAL were typically warmer and moister than during SHEBA (Fig. 8), plausibly due to }{the relatively warm West Spitsbergen Current (Aagaard et al., 1987) and/or the more synoptic active Arctic North Atlantic sector of ACLOUD/PASCAL (Serreze et al., 1997).*

SHEBA, AOE-2001 and ASCOS comparisons (P20 L1-4)
Again, the comparison here is irrelevant because of the different measurement heights and season.

Based on your suggestion, the first two sentences in the 7th paragraph in Sect. 5.4 are deleted. This paragraph now reads:
*Similar to the AOE-96, SHEBA, AOE-2001, and ASCOS campaigns (Tjernström et al., 2012), we observed inversions and these mostly in the lowest kilometer in almost all profiles when Polarstern was located in the sea ice-covered area (Fig. 5). Of the three mean profiles in Fig. 8, NP (i.e., the last and most representative key period) corresponds best to the profiles from AOE-96, SHEBA, AOE-2001, and ASCOS (Figs. 15a and 15b in Tjernström et al., 2012).*

Sea ice drift vs. atmospheric circulation (P21 L1 + P21 L2)
Change to "The sea ice drift resulting from the large-scale atmospheric circulation caused…" Delete "respectively".

Based on your suggestion, we have rewritten the 3rd paragraph in Sect. 6, which now reads:
*Relative to the long-term averages, we identified three key periods representative of the distinct synoptic states during the ACLOUD/PASCAL measurement period: (1) a cold period (CP; May 23–29; 7 days), (2) a warm period (WP; May 30 – June 12; 14 days), and (3) a normal period (NP; June 13–26; 14 days). These were characterized by (1) cold and dry Arctic air advected from the north, (2) warm and moist maritime air transported from the south and east, and (3) close-to-average temperate and moist air from a mixture of regions (but dominated by adiabatically warmed air from the west). The sea ice drift during*

*ACLOUD/PASCAL was strongly influenced by the large-scale atmospheric circulation and featured* an anomalous southerly sea ice edge in the Fram Strait, packing of the ice edge and opening of the Northeast Water Polynya *in CP, WP, and NP*, respectively. Associated with the cold and dry Arctic air flow, low-level stratus clouds prevailed over the open ocean in CP, while the warm air advections *coincided* with *complex cloud systems having considerable vertical extent* in WP. NP showed a mix of both conditions. Thus, relative to the long-term observations, we found short-term variability in atmospheric circulation *to dominate the weather condition during ACLOUD/PASCAL.*

Figure 2a
The 850hPa windspeed presented as bar graph seems a bit odd, perhaps would be better as just points or line?

We acknowledge that the time series of 850 hPa wind in Fig. 2a was plotted unconventionally. Hence, in line with more commonly used wind plotting, as well as with Figs. 4c and 4d, we have re-plotted 850 hPa wind direction and speed data in Fig. 2a as wind barbs. Doing so, we also realized that our previous 850 hPa wind directions were plotted erroneously. We are thus grateful for your suggestion of changing the wind plotting, which made us aware of and able to correct this error. With these updates, we are confident that the 850 hPa wind data above Ny-Ålesund are better depicted in the manuscript.

Furthermore, following the journal guidelines, we have added legends to each panel in Fig. 2, one in Figs. 2a and 2b and two in Fig. 2c. Additionally, we have plotted Fig. 2b on the same double y-axis format as Figs. 2a and 2c for consistency.

**(a)** *Near-surface pressure and 850-hPa wind*

[Figure]

**(b)** *Near-surface temperature and snow melt season*

**(c)** *Integrated water vapor and precipitation*

**Figure 2:** *(a) Near-surface pressure (graphs) and 850 hPa horizontal wind (bars for speed, vectors for direction), (b) near-surface air temperature (graphs) and snow melt season (solid vertical lines), and (c) vertically integrated water vapor (graphs) and precipitation (bars) measured at Ny-Ålesund (NYA; blue and black) and Polarstern (PS; red) over the*

*ACLOUD/PASCAL measurement period May 23 – June 26, 2017. Dots and intervals indicate daily average and standard deviation, respectively, over the Ny-Ålesund long-term period 1993–2016. Dashed vertical lines distinguish the Polarstern ocean-crossing periods from the ice-attached period (June 6–16).*

Figures 4 and 5
Perhaps these figures could be merged?

Please see Lifted inversions discussion (P9 L14-31) above.

Figure 13a (previous Fig. 12a)
The color scheme should have a smooth transition from day 80 to 160.

Based on your suggestions, we have re-plotted Fig. 13 (previous Fig. 12) with a more natural transitioning color scheme. We kept the four shades of each color (from dark to light) as we argue this allows the reader to more easily identify the various snow melt onset dates of each grid point.

[Figure]

**Figure 13:** *(a) Climatology (1979–2016) and (b) anomaly relative to the climatology (2017 minus 1979–2016) of snow melt onset date based on NASA GSFC data. In (b), white shading south of the 2017 sea ice edge (line) indicates open water.*

Figure 14
could be eliminated.

Please see Figure 14 (previous Fig. 13) deletion (P18 L20) above.

Figure 3 (previous Fig. A1)
The grey contours cannot be seen, even when zoomed in a lot. Perhaps just choose the 2 parameters to show here, instead of 3.

Please see Synoptic description (P7 L2 + P7 L3-7 + P7 L8 + P7 L10 + P7 L13) above.

Figures A3 and A4
could be eliminated.

Please see Figures A3 and A4 deletion (P13 L26) above.

**Response to the review by Anonymous Referee #2**

Introduction (1.)

Introduction is good from the points of view of the Arctic climate system, climate modelling, small-scale physical processes, and field campaigns. However, much more attention should be put on synoptic-scale meteorology in the study region. What is known and what are scientific challenges in the field?

We acknowledge that Sect. 1 currently does not discuss findings and uncertainties of the Arctic synoptic-scale meteorology in much detail. Instead, it presents the corresponding aspects of the Arctic climate system, climate modeling, small-scale physical processes, and field campaigns addressing these, which also are issues for predicting the synoptic-scale setting.

Nonetheless, we realize that the submitted title of the manuscript might have been misleading and have thus changed this in accordance with the manuscript focus (cf. Synoptic conditions (2.) below). The manuscript is primarily meant to provide background synoptic information for the measurements during the ACLOUD/PASCAL campaign. It is neither meant solve scientific challenges on synoptic-scale meteorology in the ACLOUD/PASCAL region nor to be a review paper on this area.

To better clarify the focus of the manuscript, we have modified the 6th paragraph in Sect. 1, which now reads:

*The intra- and interannual variability of the Arctic atmosphere is an important aspect. Therefore, it is crucial to put the short-term campaign observations into a climatological context, also to understand how representative these are. Accordingly, this paper characterizes the synoptic-scale weather and sea ice conditions during ACLOUD/PASCAL and compares them with existing climatology and other Arctic field campaigns. In doing so, the findings presented here shows how the synoptic variability is related to the variability in surface observations, atmospheric profiles, and circulation indices using ACLOUD/PASCAL background data, as well as Ny-Ålesund observations, reanalysis, operational analysis, and satellite data. The paper aims to help interpreting the upcoming detailed process studies of clouds, aerosols, energy fluxes, and other parameters observed during ACLOUD/PASCAL. Moreover, our detailed analysis gives useful insight into the processes during a typical transition period from freezing to melting conditions in the region around Svalbard. An improved understanding of processes in this region is important due to its particularly marked climate changes. Those involve an observed surface and atmospheric warming and moistening, as well as changes in the atmospheric circulation with less (more) frequent atmospheric flow from the south in summer (autumn and winter) (Maturilli and Kayser, 2017).*

Synoptic conditions (2.)

The manuscript is entitled "Synoptic development during ACLOUD/PASCAL field campaign near Svalbard in spring 2017", but there is, in fact, very little attention to synoptic-scale dynamics. With the present title, a reader expects to learn, among others, about the mechanisms (such as baroclinic instability) affecting cyclogenesis and cyclolysis during the

study period, and what was the role of the jet stream in steering the cyclone tracks. For example, on page 7, a careful description of the evolution of the synoptic situation is given, nicely linking synoptic conditions and point measurements during the campaign, but there is no deeper analysis on why the synoptic conditions developed as they did.

We thank you for pointing out the misleading title in comparison to the focus of the manuscript, as explained in more detail under Introduction (1.) above. A deeper exploration of the mechanisms behind the actual synoptic patterns would be beyond the scope of this more descriptive study. As the manuscript is meant to provide an overview of the weather conditions and air masses observed during the ACLOUD/PASCAL campaign instead and not is meant to focus on synoptic-scale dynamics, we have renamed the title to *Meteorological conditions during the ACLOUD/PASCAL field campaign near Svalbard in early summer 2017*. With this name change, we feel that the title is better reflecting the motivation for the manuscript.

Furthermore, we have moved Fig. 3 (previous Fig. A1) from the appendix to Sect. 3.1, integrated it stronger in the synoptic description there and improved this description (cf. Synoptic description (P7 L2 + P7 L3-7 + P7 L8 + P7 L10 + P7 L13) above). Hence, we believe this has improved the explanation on how and why the actual synoptic conditions (e.g., occurrences of MCAO and WAAs) developed with regards to the location and strength of the relevant centers of action (controlling low- and high-pressure systems).

Finally, we have added a paragraph at the end of Sect. 3.1 that explains why there is no analysis of synoptic-scale dynamics related to cyclones in the manuscript. This reads:

*In contrast to the N-ICE2015 expedition (Cohen et al, 2017), no prominent cyclones were observed during the ACLOUD/PASCAL campaign. Only on June 28 (indicated by the negative tendency in surface pressure in Ny-Ålesund on June 27 in Fig. 2a) a cyclone passed the region and prevented any flight activities. Hence, analysis of synoptic-scale dynamics related to cyclones similar to, for example Knudsen et al. (2015), Akperov et al. (2018) or Zahn et al. (2018), is not needed in this paper.*

Red thread (3.)
The results of the different analysis methods applied are not well linked to each other. I appreciate the analyses on in-situ and satellite observations, weather classification, air-mass distribution, atmospheric circulation and thermodynamics, clouds, and sea ice dynamics. However, the results should be put better together to summarize the synoptic development during the study period.

We gratefully acknowledge that the different analysis in the manuscript could be better linked. Based on your suggestion, we have therefore gone through the manuscript once more and tried linking the various sections better together.

In particular, we have rewritten the 1st paragraph in Sect. 3.3, which now reads:

*As shown by the observed time series, the weather during ACLOUD/PASCAL was influenced by different synoptic atmospheric patterns. A way to quantify the dominant synoptic pattern is to analyze the occurrences of MCAOs. Following Papritz et al. (2015) and*

*Kolstad (2017), the MCAO index is defined as difference between surface and 850 hPa potential temperature of each grid point, area-averaged over the eastern Greenland Sea (here defined 75.00–80.25° N, 4.50–10.50° E). Land grid cells and cells for which the surface temperature is lower than 271.5 K are* excluded *from the area averaging.*

Similarly, we have rewritten the 1st paragraph in Sect. 4.1, which now reads:

> *To assess differences in the air mass histories of the three key periods defined above we compare their mean trajectories. This analysis was performed using FLEXPART in backward mode, with input data from ECMWF operational analysis (cf. Sect. 2.4). In addition to the temporal means of PES over each key period, Fig. 7 also shows the daily center of mass trajectories of the respective key period.*

Moreover, we have rewritten the 1st paragraph in Sect. 4.2, which now reads:

> *Figure 9 complements Figs. 7 and 8 by picturing the contrasting atmospheric circulation, temperature, and humidity of the three key periods based on ERA-I data. Here, Figs. 9a, 9c, and 9e illustrate the 700 hPa geopotential height and horizontal wind of CP, WP, and NP, respectively, while the relative temperature and humidity of these periods are depicted in Figs. 9b, 9d, and 9f. In addition to their climatology, each panel depicts the anomalous conditions of the three key periods compared to their respective climatology. A more detailed evolution of the atmospheric circulation and thermodynamics observed during ACLOUD/PASCAL is presented by daily fields of these measures in Videos S1 and S2 in the supplementary material. The 700 hPa level represents the main flight level during ACLOUD (Wendisch et al., 2018).*

Furthermore, we have rewritten the 1st paragraph in Sect. 5.1, which now reads:

> *The large-scale atmospheric circulation indices Arctic Oscillation (AO; Thompson and Wallace, 1998) and Arctic Dipole (AD; Wu et al, 2006; Wang et al., 2009) represent the first and second leading empirical orthogonal function (EOF) modes of the daily 1000 hPa geopotential height anomalies poleward of 20° N and 70° N, respectively, normalized by the standard deviation of the monthly index. Another important circulation pattern in the Northern Hemisphere is the North Atlantic Oscillation (NAO), which is characterized by a pronounced north-south dipole in sea level pressure across the North Atlantic. The NAO is in this respect very similar to the AO but without the centers of action – the Aleutian Low and the Pacific High – over the Pacific Ocean. Accordingly, AO and NAO are closely related, with NAO actually being considered the regional occurrence of the hemisphere-wide pattern of AO (Thompson and Wallace, 1998). The analysis therefore focused on AO to provide broader information on the large-scale dynamics.*

Finally, we have rewritten the 1st paragraph in Sect. 5.2, which now reads:

> *The onset of snow melt is a key parameter for Arctic amplification as it determines the seasonal change of the surface energy budget. Due to the melt of snow and later sea ice, radiative and sensible heat is efficiently stored in form of latent heat in the Arctic Ocean. The date of early snow melt onset is retrieved from passive microwave satellite observations over sea ice (Markus et al., 2009). This date represents the first day under melting conditions and is plotted* jointly for *both the climatological period and* the 2017 *deviation from* the *climatological period in Fig. 13.*

Relevance scientific community (4.)

The manuscript includes a lot of information that will certainly be useful for those working with ACLOUD/PASCAL data, but its usefulness for a broader scientific community is not equally clear. To deserve publication in a high-level peer-reviewed journal, such as ACP, the manuscript should be useful for a broader community. This challenge can probably be met via a careful concern on topics 1 to 3 above and a proper discussion on the purpose of the manuscript.

We strongly agree that the manuscript should be useful for a broader scientific community beyond the ACLOUD/PASCAL community. In this regard, we would particularly like to highlight Sect. 5, which offer extraordinary analysis compared to other meteorological papers from Arctic field campaigns. In this section, the anomaly of the ACLOUD/PASCAL campaign is discussed in the context of its large-scale region, its climatology, and in comparison to similar campaigns.

Data from the ACLOUD/PASCAL campaign will be made available through the PANGAEA Data Publisher (https://www.pangaea.de/). For a holistic interpretation of these, our manuscript will be essential to understand where the air masses measured were coming from. Similarly, current studies in preparations from ACLOUD/PASCAL depend on a manuscript providing an overview of the meteorological conditions during the campaign without the usage of the measured data from the campaign. These studies will be part of the same special issue in ACP as our manuscript is submitted to, *Arctic mixed-phase clouds as studied during the ACLOUD/PASCAL campaigns in the framework of (AC)³*. Hence, our manuscript is not a stand-alone manuscript.

For this reason, we would like to stress that we are grateful for your comments, which have helped us addressing the shortcomings of our study. With the changes to the manuscript explained in this response, we believe that we have taken significant steps in making the manuscript more useful and interesting to the broader scientific community.

ACLOUD/PASCAL season (Title)

Consider replacing spring by early summer.

Based on your suggestion, we have renamed the title to *Meteorological conditions during the ACLOUD/PASCAL field campaign near Svalbard in early summer 2017*.

As a result, the 2nd paragraph in Sect. 3.3 now reads:

*Time series of the 6-hourly MCAO index are calculated for the ACLOUD/PASCAL period and used to identify events of cold air outbreaks. A new event begins when the index is greater than 0 K and ends if the index falls below 0 K. Then, the last time for which the MCAO index > 0 K is set as the final time step of the event. Events are recorded only if an index value of at least 2 K is reached and the duration is at least 48 hours. The maximum MCAO index of each event is required to occur within the ACLOUD/PASCAL measurement period, but the events are allowed to start any time in May or by the end of June. The threshold of 2 K is lower than in studies focusing on the cold season (e.g., 3 K in Kolstad, 2017). The lowered threshold accounts*

*for the fact that MCAOs occur considerably less frequent and are considerably less severe in early summer than in winter (Fletcher et al., 2016).*

Similarly, the last paragraph in Sect. 5.3 now reads:

*Warm air advections are more common in early summer, with 21 events recognized over the ACLOUD/PASCAL comparison period (Fig. 14b). Duration and strengths of these reached up to 12 days and 14 K, respectively, although the majority lasted less than 8 days and were weaker than 9 K. In 2017, two moderate WAAs took place (cf. Fig. 6). These lasted 6 and 7 days and had intensities of 9.1 K to 10.3 K, respectively.*

WAA events + Westerly air (P1 L10 + P1 L11)
. . . two cases of warm-air advection.
What is "westerly air"?

Based on your suggestion, we have rewritten the abstract, which now reads:

*The two concerted field campaigns Arctic CLoud Observations Using airborne measurements during polar Day (ACLOUD) and the Physical feedbacks of Arctic planetary boundary level Sea ice, Cloud and AerosoL (PASCAL) took place near Svalbard from May 23 to June 26, 2017. They were focused on studying Arctic mixed-phase clouds and involved observations from two airplanes (ACLOUD), an icebreaker (PASCAL), a tethered balloon, as well as ground-based stations. Here, we present the synoptic development during the 35 day period of the campaigns, using near-surface and upper-air meteorological observations, as well as operational satellite, analysis, and reanalysis data. Over the campaign period, short-term synoptic variability was substantial, dominating over the seasonal cycle. During the first campaign week, cold and dry Arctic air from the north persisted, with a distinct but seasonally unusual cold air outbreak. Cloudy conditions with mostly low-level clouds prevailed. The subsequent two weeks were characterized by warm and moist maritime air from the south and east, which included two events of warm air advection. These synoptical disturbances caused lower cloud cover fractions and higher-reaching cloud systems. In the final two weeks, adiabatically warmed air from the west dominated, with cloud properties strongly varying in between the range of the two other periods. Results presented here provide synoptic information needed to analyze and interpret data of upcoming studies from ACLOUD/PASCAL, while also offering unprecedented measurements in a sparsely observed region.*

Pithan and Mauritsen (2014) (P2 L6)
Consider referring to Pithan and Mauritsen (2014)

Based on your suggestion, we have added a reference to Pithan and Mauritsen (2014) to the 1st paragraph in Sect. 1, which now reads:

*The phenomenon of Arctic amplification – the 2–3 times higher warming of the Arctic relative to the global atmosphere – is a major indication of current drastic Arctic climate changes (Serreze and Barry, 2011). A number of potential causes for this special feature of the Arctic climate system are discussed, which include various interconnected processes and feedback mechanisms, such as sea ice loss and surface albedo feedback, meridional atmospheric and oceanic energy fluxes, and atmospheric radiation effects linked to temperature, water vapor and clouds (Pithan and Mauritsen, 2014). Still, the relative*

*importance of these different feedback mechanisms is subject of the current scientific debate (Wendisch et al., 2017).*

"Are key" vs. "are the key" (P2 L20)
are the key

Based on your suggestion, we have rewritten the 2nd paragraph in Sect. 1, which now reads:

*Climate models have difficulties in reproducing the observed drastic Arctic climate changes, and therefore the uncertainty in Arctic climate projections is larger than in other parts of the world (Stocker et al., 2013). This issue is related to major gaps in understanding of key processes particularly important for the Arctic climate system. Significant uncertainties in the parameterization of subgrid-scale processes remain one of the major challenges for realistic climate simulations, particularly in high latitudes (Vihma et al., 2014). Further important open questions are associated with cloud physical processes (e.g., Tjernström et al., 2008; Boer et al., 2014; Pithan et al., 2014) and sea ice albedo-cloud radiative interactions (e.g., Karlsson and Svensson, 2013; English et al., 2015). The results of different Arctic climate models substantially disagree; they also generally do not match with observations in particular with respect to hydrometeor phase partitioning in mixed-phase clouds (Morrison et al., 2011; McIlhattan et al., 2017) and the vertical structure of the atmospheric boundary layer (ABL; Svensson and Lindvall, 2015), which are interrelated (Lüpkes et al., 2010; Barton et al., 2014; Pithan et al., 2014). Those biases can considerably affect the water vapor and temperature profiles and the atmospheric radiation budget, which can consequently alter the individual climate feedback (Kim et al., 2016). To make substantial progress in these areas, dedicated observational campaigns in the Arctic are crucial.*

In making this change, we replaced our usage of the word "key" (as defined by Cambridge Dictionary, https://dictionary.cambridge.org/dictionary/english/key?q=the-key-to-sth, and Merrian Webster dictionary, https://www.merriam-webster.com/dictionary/key%20to) with "crucial" to avoid confusion.

"Particularly marked climate changes" (P3 L21-22)
Briefly describe the the "particularly marked climate changes".

Based on your suggestion, we have added a sentence at the end of the 6th paragraph in Sect. 1 to describe the "particularly marked climate changes". This paragraph now reads:

*The intra- and interannual variability of the Arctic atmosphere is an important aspect. Therefore, it is crucial to put the short-term campaign observations into a climatological context, also to understand how representative these are. Accordingly, this paper characterizes the synoptic-scale weather and sea ice conditions during ACLOUD/PASCAL and compares them with existing climatology and other Arctic field campaigns. In doing so, the findings presented here shows how the synoptic variability is related to the variability in surface observations, atmospheric profiles, and circulation indices using ACLOUD/PASCAL background data, as well as Ny-Ålesund observations, reanalysis, operational analysis, and satellite data. The paper aims to help interpreting the upcoming detailed process studies of clouds, aerosols, energy fluxes, and other parameters observed during ACLOUD/PASCAL. Moreover, our detailed analysis gives useful insight into the processes during a typical*

*transition period from freezing to melting conditions in the region around Svalbard. An improved understanding of processes in this region is important due to its particularly marked climate changes. Those involve an observed surface and atmospheric warming and moistening, as well as changes in the atmospheric circulation with less (more) frequent atmospheric flow from the south in summer (autumn and winter) (Maturilli and Kayser, 2017).*

Sea ice thickness (Sect. 2.3)
were any data collected on sea ice thickness?

For the manuscript, no sea ice thickness data were collected. However, as part of the PASCAL field campaign, measurements on sea ice thickness were made (for more information, please see Macke and Flores, 2018).

Clarification analysis data + Use of forecasts (P6 L7 + P6 L19 + P6 L9-10)
. . . reanalyses and operational analysis data . . .
Unclear sentence
ECMWF operational analysis

Based on your suggestion, we have rewritten the 1st and last paragraphs in Sect. 2.4, which now read:

*Because in situ and satellite data can only provide a limited perspective, reanalysis and operational analysis data from the European Centre for Medium-Range Weather Forecast (ECMWF) are used to best describe the state of the atmosphere over the broader domain and longer time scales. As one of the objectives of ACLOUD/PASCAL is to investigate the skills of forecast models, explicitly no forecasts are analyzed in this manuscript.*

*ECMWF operational analysis data were obtained on a 0.25° x 0.25° horizontal grid. These were used for the synoptic description in Sect. 3.1, as well as provided the input for the Lagrangian particle dispersion model Flexible Particle Dispersion (FLEXPART; Stohl et al., 2005) used to analyze the history of air masses arriving in Ny-Ålesund in Sect. 4.1.*

Clarification surface cooling in association with WAA (P8 L3)
Clarify the sentence with ". . . surface cooling during warm-air advection ...".

Based on your suggestion, we have rewritten the relevant sentence in 8th paragraph in Sect. 3.1, which now reads:

*June 6 was also the date when the observations from the ice-attached Polarstern started. Over its first days in the ice, the sea ice camp observed an increase in near-surface pressure due to a high pressure ridge east of Svalbard (Fig. 3d), reaching a maximum of 1029 hPa on June 8. IWV from 6 kg m⁻² to 17 kg m⁻² on June 9 (Fig. 2c) and in near-surface air temperature from -8° C to +2° C on June 10 (Fig. 2b). In other words, the above-freezing temperature on Polarstern while surrounded by sea ice (June 1–17) occurred four days after that in Ny-Ålesund, which is later than that arising from the pure air mass transport. This delay can be explained by the more northerly location of Polarstern within the compact sea ice, where surface cooling fosters a stable inversion layer close to the ground while warm air advection occurs in the free troposphere above. As long as the inversion is not destroyed, it*

*remains cold at the lowest levels. Anomalously warm and moist air was also observed in Ny-Ålesund these days, but with less intense changes due to the already warm and moist air since June 6. Thus, while the synoptic conditions were similar for Ny-Ålesund and Polarstern during June 6–8 (Fig. 2), local factors (e.g., sea ice distribution) probably played an important role for the difference between the two stations at about 335 km apart.*

Objectively vs. subjectively (P9 L19)
objectively or subjectively chosen?

Please see Lifted inversions discussion (P9 L14-31) above.

PES description (P11 L26)
Briefly describe what is the potential emission sensitivity.

Please see Backward air mass trajectory analysis and discussion (P11-12) above.

Adiabatic warming vs. cooling (P12)
If airmass flows over Spitsbergen or Greenland, it first experiences adiabatic coolinf during the ascent, and then adiabatic warming during the descent. Your attention seem to be resctricted to the latter. Why? Further, do you catch the true adiabatic warming/cooling effects, if you integrate PES in the vertical?

The true adiabatic warming/cooling effects indeed do not directly follow from the vertically integrated PES values. This was instead revealed in measurements, indicating that relatively warm and humid air masses reached Ny-Ålesund, which could be linked with the trajectories coming from the Greenland ice sheet or over Spitsbergen. Hence, the relevant discussion is a plausible explanation of what happened. To clarify this uncertainty, we have adapted the relevant paragraph slightly.

For this reason, we have rewritten the last two sentences in the 5th paragraph in Sect. 4.1, which now reads:
*The PES distribution of the last key period – NP – was a mixture of the two former key periods. Most of the Arctic Ocean and the Nordic Seas were then sources of air mass origin, but the highest density was found in air arriving Ny-Ålesund from the west (Fig. 7c). The relatively average temperate and humid air observed here (Figs. 2b, 2c, 3a, and 3b) can potentially result from the air masses passing over the sea ice north of Greenland, the open ocean south of Svalbard or the Greenland ice sheet. These air masses could be heated either by adiabatic motions or through sensible or latent heat fluxes from the ocean into the atmosphere during their transport from the sea ice/open ocean transition zone in the Fram Strait to Ny-Ålesund.*

Radiosonde entering the free troposphere (P12 L26-28) + WP profiles in the lowest 300 hPa (P13 L8-12)
Explain better. All radiosondes enter the free troposphere whether there are mountains or not.
The paragraph is very unclear. Clarify what is the effect of sea ice on the specific humidity in the lowermost 300 hPa.

Please see Figure 8 (previous Fig. 7) discussion (P12-13) above.

Chan and Comiso (2013) (P15 L15)
over the open ocean?

Based on your suggestion, we have rewritten the relevant sentence in 3rd paragraph in Sect. 4.4, which now reads:

*Of the three key periods, the highest cloud cover fraction is observed during CP, with an average of about 85 % in the central ACLOUD/PASCAL region (Fig. 11a). In general, the highest cloud cover is observed over the open ocean (cf. Fig. 10a). This is in agreement with the results by Chan and Comiso (2013), who found a cloud cover fraction of about 88 % over* *open water* *across the whole Arctic and all seasons.*

Fog conditions (Sect. 4.4)
Add brief characterization on fog conditions.

Based on your suggestion, we have added a paragraph to Sect. 4.4, which characterizes the fog conditions during ACLOUD/PASCAL. It reads:

*Similarly, a more complete picture of fog conditions will be made possible from the analysis of the wealth of ground and airborne remote sensing observations during ACLOUD/PASCAL. It is not possible to infer fog conditions from the satellite observations as a high cloud top pressure could either be related to low stratus or high fog conditions. Furthermore, due to the strong topographical influence on their location, observations from Ny-Ålesund are not representative for the ACLOUD/PASCAL region. The ice-attached Polarstern had a more representative location, from which visual observations are available. Here, fog was observed into the days of June 6 and 8, as well as on June 12. However, the visibility was mostly around 5 km and never fell below 500 m, indicating that low-hanging stratus clouds rather than fog was present most of the time.*

AO discussion (P17)
Better explain the association with colder and wetter conditions in mid- latitudes. Which season you refer to? The all paragraph on AO makes more sense for winter than summer. In winter, Arctic cold-air outbreaks are typically associated with cold and dry conditions in mid-latitudes.

We thank you for pointing out that the discussion of AO was too general and lacked seasonal specifics. To mitigate this, we have rewritten the 2nd (previous 3rd) paragraph in Sect. 5.1 and complemented it with more references:

*AO and AD are measures of the zonal and meridional wind patterns. AO describes the variability in the strength of the polar vortex. A positive AO index is associated with a lower-than-average pressure over the Arctic, a strong polar vortex, and a mainly zonal jet structure. Cold polar air mass is therefore more confined and located further poleward. In contrast, a negative AO index is linked to higher-than-average pressure over the Arctic, a weaker vortex, and a stronger meridional component of the jet stream. As a result, positive AO indices correlate with more numerous and deeper cyclones in the Arctic region,* *with storm tracks being shifted to the north (Simmonds et al., 2008). Conversely,* *negative indices are associated*

*with more frequent blocking high events and persistent weather conditions, as well as with more likely MCAO events mainly in winter and spring (Overland et al., 2015). Toward summer, the AO pattern is displaced further northward and the meridional extent of its signal is considerably reduced (Ogi et al., 2004). A negative AO circulation in summer is nevertheless still supposed to cause substantial surface and tropospheric cooling and enhanced precipitation in midlatitudes (e.g., Hufeng and Feng, 2010; Wu et al., 2016).*

AD discussion (Sect. 5.1)
I wonder if AD, as defined by Wu et al. (2006) and Wang et al. (2009), is the best metrics to characterize meridional circulation patterns and atmospheric forcing on sea ice drift in the Arctic. See Vihma et al. (2012, GRL) for various weaknesses of AD. The Meridional Circulation Index (Francis and Vavrus, 2015, Env. Res. Lett.) may be a much more relevant metrics. Note that it can also be calculated on the basis of mean-sea-level pressure.

We thank you for pointing out that the interpretation of the AD index is not always straightforward for reasons discussed by Vihma et al. (2012). In this study, the AD index was, however, found to be the best metric by correlation to the Arctic summer sea ice drift speed in the Fram Strait, which is the focus of this manuscript.

Based on these aspects, we have added a 4th paragraph to Sect. 5.1 that briefly discusses the limitations of the AD and its usage. This reads:
*However, the connection between AD and Arctic sea ice drift is not always straightforward since the pressure pattern affecting AD may be oriented off the direction of the Transpolar Drift Stream, as pointed out by Overland and Wang (2010). Furthermore, the AD index is sensitive to the time period and geographical area considered in the calculation and is also dependent on the reanalysis data used. Meridional circulation indices based on the mean sea level pressure gradient across the Fram Strait or the Transpolar Drift Stream can provide a better quantitative relationship between the atmospheric forcing and sea ice drift speed throughout the year. Nevertheless, in summer, when the axis of the AD pattern is usually oriented along the Fram Strait, the AD index is found to correlate well with the sea ice evolution in the Fram Strait/Svalbard area (Vihma et al., 2012), which is the focus of the following qualitative analysis.*

Additionally, following Vihma et al. (2012), we calculated the simple Central Arctic Index (CAI), which supports the qualitative relationship between AD and the sea ice evolution presented so far. Due to its good agreement to the AD index and in order to not overload the manuscript with another index, we prefer not to include the CAI analysis in the revised manuscript.

Atmospheric circulation short-term variability vs. long-term forcing (P21 L4-5)
The sentence could be clarified. How would you characterize the long-term background forcing of the Arctic amplification on atmospheric circulation? It is fairly trivial that short-term variability in circulation is stronger than this forcing, which is not yet well known.

We acknowledge that the relevant sentence was badly worded. Hence, in the updated manuscript we have clarified it.

As a result, the 3ʳᵈ paragraph in Sect. 6 now reads:

*Relative to the long-term averages, we identified three key periods representative of the distinct synoptic states during the ACLOUD/PASCAL measurement period: (1) a cold period (CP; May 23–29; 7 days), (2) a warm period (WP; May 30 – June 12; 14 days), and (3) a normal period (NP; June 13–26; 14 days). These were characterized by (1) cold and dry Arctic air advected from the north, (2) warm and moist maritime air transported from the south and east, and (3) close-to-average temperate and moist air from a mixture of regions (but dominated by adiabatically warmed air from the west). The sea ice drift during ACLOUD/PASCAL was strongly influenced by the large-scale atmospheric circulation and featured an anomalous southerly sea ice edge in the Fram Strait, packing of the ice edge and opening of the Northeast Water Polynya in CP, WP, and NP, respectively. Associated with the cold and dry Arctic air flow, low-level stratus clouds prevailed over the open ocean in CP, while the warm air advections coincided with complex cloud systems having considerable vertical extent in WP. NP showed a mix of both conditions. Thus, relative to the long-term observations, we found short-term variability in atmospheric circulation to dominate the weather condition during ACLOUD/PASCAL.*

Synoptic forecasting in climate models (P21 L17-18)
What do you meant by synoptic forecasting in climate models?

We acknowledge that the relevant reference to synoptic forecasting in climate models was badly worded. Hence, in the updated manuscript we have removed it.

As a result, the final paragraph in Sect. 6 now reads:

*Our focus was limited to the North Atlantic sector of the Arctic. Hence, the results presented here do not necessarily translate to the entire Arctic climate system because the regional differences are too large (e.g., Serreze et al., 2011; Cavalieri and Parkinson, 2012; Koyama et al., 2017). For example, sea ice coverage in the region was anomalously high and reached far south as a result of the strong southward drift during CP and, albeit weaker, still southward drifts during WP and NP. Nevertheless, considering the sparsely observed Arctic region, the extensive ACLOUD/PASCAL campaign offers unique measurements covering the entire tropospheric column, with observations over the open ocean, sea ice, and snow. Most measurements performed during ACLOUD/PASCAL will be continued in the framework of MOSAiC, including a one-year ice drift of Polarstern and numerous aircraft- and ground-based activities. Thus, while MOSAiC will strongly benefit from the results and experiences gained from ACLOUD/PASCAL, the continuity of observations in this Arctic region is anticipated to considerably improve the understanding of the cloud-related processes in the Arctic atmosphere, as well as the ocean-ice-atmosphere interaction from turbulent and radiative energy fluxes. Ultimately, this will strengthen synoptic forecasting in weather models, benefiting actors beyond the scientific community.*